# VideoAgentTrek: Computer Use Pretraining from Unlabeled Videos

Dunjie Lu[1,2*], Yiheng Xu[1,2*], Junli Wang[2*], Haoyuan Wu[1], Xinyuan Wang[1], Zekun Wang[2], Junlin Yang[1], Hongjin Su[1], Jixuan Chen[1], Junda Chen[1], Yuchen Mao[1], Jingren Zhou[2], Junyang Lin[2], Binyuan Hui[2†], Tao Yu[1†]

[1]The University of Hong Kong      [2]Qwen Team, Alibaba Group

## Abstract

Training computer-use agents requires massive amounts of GUI interaction data, but manually annotating action trajectories at scale is prohibitively expensive. We present VideoAgentTrek, a scalable pipeline that automatically mines training data from publicly available screen-recorded videos, eliminating the need for manual annotation. Our approach addresses a key challenge: raw videos contain implicit demonstrations but lack explicit action labels. To solve this, we develop Video2Action, an inverse dynamics module (IDM) with two components: (1) a video grounding model that detects and localizes GUI actions with precise temporal boundaries, and (2) an action-content recognizer that extracts structured parameters like click coordinates and typed text. Applied to 39,000 YouTube tutorial videos, our pipeline generates 1.52 million interaction steps. We leverage this data through continued pretraining followed by supervised fine-tuning. On OSWorld-Verified, our approach improves task success rates from 9.3% (SFT-only baseline) to 15.8%, a 70% relative improvement. On AgentNetBench, step accuracy increases from 64.1% to 69.3%. Our results demonstrate that passive internet videos can be transformed into high-quality supervision for computer-use agents, providing a scalable alternative to expensive manual annotation.

## 1 Introduction

Teaching machines to use computers like humans do (clicking buttons, typing text, navigating interfaces) represents a fundamental challenge in AI. While recent advances in vision-language models have made computer-use agents increasingly feasible (Bai et al., 2025; Qin et al., 2025; Team et al., 2025; Wang et al., 2025b), their development remains bottlenecked by data availability. Training these agents requires extensive trajectories that precisely document GUI interactions: screenshots paired with exact action parameters like click coordinates $(x, y)$ and typed strings. However, creating such datasets through manual annotation is extraordinarily expensive, making it impractical to achieve the scale necessary for robust generalization across diverse applications and operating systems.

Meanwhile, the internet hosts millions of screen-recorded tutorials where humans demonstrate computer use, from Excel tutorials to software walkthroughs. These videos implicitly contain the supervision we need: they show where users click, what they type, and how interfaces respond. Yet this resource remains untapped because videos lack the structured action labels required for training. The cursor movements are visible but not tracked; the typed text appears but isn't extracted; the timing of actions is implicit but not annotated. We can learn to automatically extract structured action trajectories from raw videos by training specialized models to detect *when* actions occur and infer *what* their parameters are, effectively converting passive recordings into active training data.

We introduce **VideoAgentTrek**, a scalable pipeline that mines computer-use trajectories from publicly available unlabeled videos without manual annotation. Our approach employs Video2Action, an inverse dynamics module (IDM) with two stages: First, an action event detection model performs dense event detection, identifying action types and their precise temporal boundaries (e.g., `click` at $[1.5, 2.0]$s, `type` at $[3.5, 5.5]$s). Second, the action parameterization model, an action-content

---

*Equal contribution. † Corresponding authors. Page: https://videoagenttrek.github.io

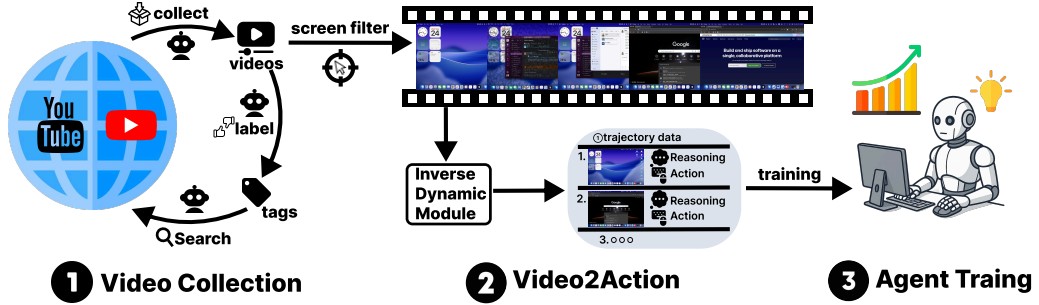

Figure 1: Overview of VIDEOAGENTTREK. (1) **Video Collection**: crawl screen-recorded tutorials and filter GUI footage with SCREENFILTER. (2) **Video2Action**: an inverse dynamics module that first performs dense action-event detection to localize clips and assign action types, then *action parameterization* (e.g., click coordinates, typed text) to yield structured (screenshot, action, parameters) trajectories. (3) **Agent Training**: use the mined trajectories for continued pretraining and supervised finetuning of computer-use agents.

recognizer, analyzes these localized segments to extract structured parameters (pointer coordinates for clicks, literal text for typing), yielding complete $(screenshot, action, parameters)$ trajectories suitable for training.

VIDEOAGENTTREK enables large-scale computer-use pretraining with unlabeled web videos. From 39,000 YouTube videos, we automatically extract 1.52 million interaction steps. This represents not just more data, but more diverse data: the trajectories span hundreds of applications across Windows, macOS, and web platforms, capturing interaction patterns that would be infeasible to annotate manually.

We validate VIDEOAGENTTREK with a two-stage training recipe: continued pretraining on the mined trajectories followed by supervised fine-tuning on a curated dataset. This combination leverages the broad coverage from videos to learn fundamental GUI interaction patterns, while supervised fine-tuning sharpens task-specific performance. Our models achieve 15.8% task success on OSWorld-Verified compared to 9.3% for baselines, a 70% relative improvement. The gains are particularly pronounced in online environments where robustness to visual variation matters most. We summarize our main contributions and findings below:

- We propose VIDEOAGENTTREK, an unsupervised approach to training computer-use agents that automatically converts screen-recorded videos into structured training data through learned inverse dynamics, thereby eliminating the need for manual annotation.

- Our VIDEO2ACTION module implements inverse dynamics, combining action event detection with millisecond-precision temporal localization and action parameter extraction. It enables accurate reconstruction of GUI interactions (clicks, typing,...) from raw video without ground-truth labels.

- Experiments demonstrate that our approach achieves 15.8% task success on OSWorld-Verified compared to 9.3% for SFT-only baselines (70% relative improvement), and improves step accuracy on AgentNetBench from 64.1% to 69.3%, validating that passive internet videos can provide effective supervision at scale.

- We provide a reproducible pipeline and training methodology that enables researchers to leverage publicly available screen recordings for computer-use agent training. To facilitate future research, we release SCREENFILTER for efficient GUI filtering and VIDEO2ACTION for action extraction as open-source tools.

## 2 VIDEOAGENTTREK

We introduce **VIDEOAGENTTREK**, a video-driven pipeline that turns web tutorials into training supervision for computer-use agents. Each trajectory is a sequence $\mathcal{R} = \{(I_k, r_k, a_k, \pi_k)\}_{k=1}^{K}$ following Yao et al. (2023), where $I_k$ is a representative screenshot, $r_k$ is a brief inner monologue,

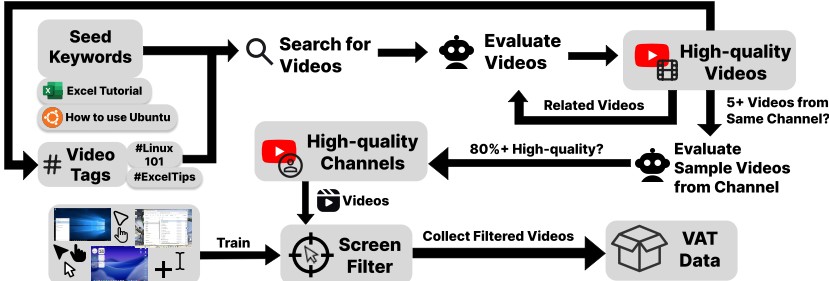

Figure 2: **Video candidate auto-discovery.** From seed keywords and tags, we search and evaluate videos, expand to related videos and high-quality channels ($\geq$80% pass), and iteratively collect GUI-containing videos for VAT.

$a_k \in \mathcal{A}$ is the action type (e.g. click, type), and $\pi_k$ is the action content (e.g., pointer $(x, y)$ or typed text). The pipeline has three parts:

- **Video collection and preprocessing.** We crawl tutorial videos with seeded queries and tag expansion, apply human-in-the-loop screening, and use cursor-based filtering to retain screen segments with GUI interactions (Section 2.1).

- **VIDEO2ACTION.** From raw video, we recover stepwise supervision without manual labels: (i) dense event detection produces typed segments with tight start/end times; (ii) action identification infers parameters $\pi_k$ (e.g., click coordinates, typed strings); and (iii) a short inner monologue $r_k$ makes the intent explicit. Assembling these per-clip steps yields ReAct tuples for training (Section 2.2).

- **Agent training.** We combine large-scale agentic data produced by the method with human demonstrations and targeted GUI grounding pairs, and train an end-to-end agent in two stages: interleaved video–text pretraining followed by instruction-style finetuning (Section 2.3).

This structure scales supervision to web-scale while preserving the stepwise semantics needed for robust computer-use policies.

## 2.1 VIDEO COLLECTION AND PREPROCESSING

### 2.1.1 VIDEO CANDIDATE AUTO-DISCOVERY

We employ a scalable pipeline for video collection that leverages channel coherence—the observation that YouTube channels typically maintain consistent content types and quality. Starting from seed keywords such as "Excel tutorial" and "How to use Windows", we validate initial results and extend to entire channels when sampling indicates high quality (i.e., when $\geq 80\%$ of sampled videos meet our criteria). This channel-based expansion enables efficient scaling: validated channels become trusted sources for candidate videos, while their tags and metadata enable iterative discovery.

When we identify high-quality channels through seed validation, we include all their videos as candidates rather than individually vetting each one. This approach deliberately optimizes recall over precision, as the subsequent SCREENFILTER stage ensures final data quality. The channel coherence property—where content creators typically focus on consistent topics—makes this expansion particularly effective.

Through iterative rounds of keyword search, channel expansion, and tag extraction, we transform a small set of manually validated seeds into 55,000 candidate videos ($\sim$10,000 hours). This process requires minimal human oversight: initial quality validation on seed videos and periodic verification of expansion effectiveness. The resulting candidate pool intentionally includes some non-GUI content (presentations, tutorials with mixed content), which our filtering stage handles efficiently.

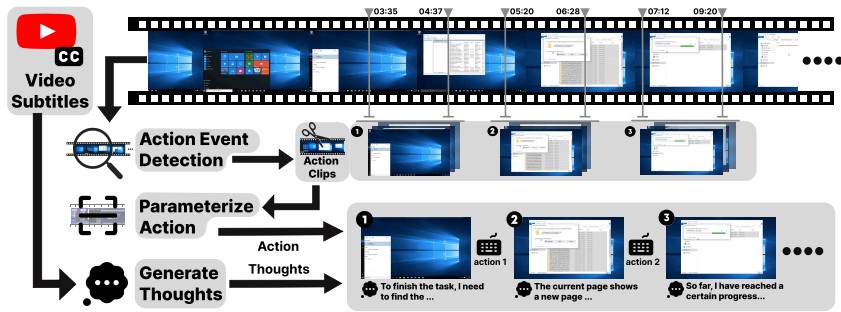

Figure 4: **Overview of VIDEO2ACTION**: Given a screen-capture video (with optional subtitles), the module (1) detects GUI action events and segments clips, (2) parameterizes each action (type and arguments), and (3) generates step-level thoughts, yielding training-ready sequences of {action clip, action, thought}

### 2.1.2 VIDEO PREPROCESSING WITH SCREENFILTER

Although keyword-based searches typically retrieve relevant computer operations, they also include non-interactive segments, such as explanatory sections where the presenter uses PowerPoint or other presentation tools. Additionally, some of the videos retrieved through this method may not meet the standards for GUI interaction content.

To address this, we developed SCREENFILTER, a lightweight cursor detection model upon YOLOv8x (Reis et al., 2024) to efficiently extract video segments that focus exclusively on GUI interactions. Using the detection results, we retain video segments where at least 80% of the frames contain a cursor for 6 seconds or more, with a 2-second merge gap for temporal smoothing. When applied to our corpus, SCREENFILTER successfully extracts 7,377 hours of verified GUI interactions from 10,000 hours of raw video. SCREENFILTER's data construction and evaluation details are in Appendix C.

### 2.1.3 VIDEOAGENTTREK DATA ANALYSIS

**Quality and relevance.** We collected 55k screen-capture videos (about 10,000 hours) from 50+ channels. The corpus is predominantly clear (about 97% are 720p or higher) and most clips are minutes long, yielding sustained, readable interactions suitable for our pipeline (Table 9). A lightweight title/description audit groups videos into tutorials, background pieces, tech talks, and unrelated; tutorials dominate (69.6%), with the remainder used mainly for tag mining or removed during filtering (Table 10). Together, these checks indicate that the collected data are both visually clean and topically aligned with computer-use supervision.

**Data classification.** We label each video as daily, office, workflow, professional, operating-system (OS), or other using a lightweight GPT-4.1 pass over the title and a short transcript snippet. The distribution (Figure 3) is skewed toward OS-level operations (∼36%), followed by professional (∼19%), daily (∼18%), and office (∼16%); workflow is smaller (∼7%) with a small remainder labeled as other (∼4%). This indicates broad coverage with a bias toward system and professional use cases.

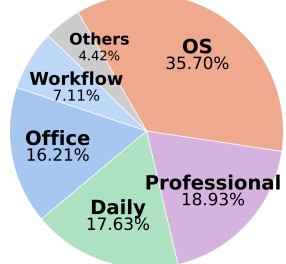

Figure 3: Domain distribution.

### 2.2 VIDEO2ACTION: INVERSE DYNAMICS MODULE

We develop VIDEO2ACTION, an inverse dynamics module (IDM) that extracts structured action supervision from unlabeled GUI videos. Following insights from robotics where inverse dynamics recovers actions from observations (Nguyen-Tuong & Peters, 2010), VIDEO2ACTION detects GUI events (clicks, drags, scrolls, typing) and infers their parameters directly from pixel changes. This yields training-ready (screenshot, action) pairs without manual annotation, forming the second core component of our VideoAgentTrek toolkit.

### 2.2.1 ACTION EVENT DETECTION

**Task.** Given an unlabeled screen-capture video $v$ (length $T$), perform prompt-free, dense event detection: predict a set of typed GUI interactions with tight temporal bounds,

$$f_\theta(v) \rightarrow \mathcal{S} = \{(a_k, t_k^{\text{s}}, t_k^{\text{e}})\}_{k=1}^K, \quad a_k \in \mathcal{A}, \ 0 \le t_k^{\text{s}} < t_k^{\text{e}} \le T.$$

Unlike query-based setups, our input contains only $v$; the output is a multi-event set with both action types and start/end timestamps.

**Approach.** We equip a VLM with video grounding so that, given a clip, it emits a sequence of $(a_k, t_k^{\text{s}}, t_k^{\text{e}})$ for all GUI actions, reframing keyframe detection as multi-class temporal event detection with tight bounds. **(1) Training data**: We utilize the annotation tool provided by OpenCUA (Wang et al., 2025b) to obtain synchronized screen videos and timestamped GUI interactions (mouse/keyboard events). These raw demonstration logs are then used to create temporal-grounding supervision, allowing precise event detection without manual annotation. **(2) Model training**: We leverage Qwen2.5-VL (Bai et al., 2025) as the base model, benefiting from its multimodal understanding and fine-grained spatiotemporal capabilities. We perform full-parameter supervised fine-tuning on the Qwen2.5-VL-7B-Instruct model to enable it to generate ordered, typed event spans directly from raw video clips. **(3) Evaluation**: We evaluate the detector in two phases. First, we check its performance on a small curated subset from the source corpus, ensuring tight boundaries and full recovery of relevant GUI actions. Second, we apply the model to unseen web tutorials and conduct blinded manual review to assess its robustness and real-world usability. Details are provided in Appendix D.

### 2.2.2 ACTION PARAMETERIZATION

**Task.** Given a detected action segment $v_k = v[t_k^{\text{s}} : t_k^{\text{e}}]$ with type $a_k \in \mathcal{A}$, predict the action content (parameters) $\pi_k$:

$$h_\phi(v_k) \rightarrow (\hat{a}_k, \pi_k).$$

For example, a click segment yields $h_\phi(v_k) \rightarrow (\text{click}, (x, y))$, while a typing segment yields $h_\phi(v_k) \rightarrow (\text{type}, \langle content \rangle)$.

**Approach.** We build a recognizer $h_\phi$ that, for each detected segment $v_k$, predicts both the action type and its parameters $(\hat{a}_k, \pi_k)$. **(1) Training data**: We start from the OpenCUA raw demonstration logs, which pair screen-capture video with timestamped mouse and keyboard events. Each event is converted into type-specific parameter labels and temporally aligned to its clip, yielding prompt-free supervision that captures the exact content of the interaction. **(2) Model training**: Using Qwen2.5-VL (7B Instruct) as the base, we perform full-parameter supervised fine-tuning so the model maps $v_k$ directly to $(\hat{a}_k, \pi_k)$; when available, we optionally condition on the detector's $a_k$ to stabilize type predictions. **(3) Evaluation**: Because ground-truth object boxes are unavailable, we evaluate only on unseen web tutorials via blinded manual review, assessing whether the predicted action type and parameters are correct and practically actionable. Details are provided in Appendix E.

### 2.2.3 INNER MONOLOGUE GENERATION

Dense event detection and action identification recover what happened on screen but omit the stepwise rationale. We therefore generate a brief inner monologue $r_k$ before each action to make explicit the intent, the local plan, and the expected state change (e.g., "type query into the search box to reveal results," "scroll to bring the 'Settings' button into view"). Explicit rationales provide structured supervision for planning and credit assignment, tie cursor–target grounding to goals and affordances, and improve robustness on long-horizon tasks via better error detection and recovery. Recent GUI-agent work that injects step-level "thoughts" or System-2 reasoning reports notable gains in perception, grounding, and task execution, motivating our inclusion of $r_k$ in ReAct-style trajectories (Xu et al., 2025b; Qin et al., 2025; Wang et al., 2025b).

We cast inner-monologue generation as conditional paraphrasing with GPT-5 Medium. For each step $k$, we build a structured prompt that includes: (i) the detected action type $a_k$; (ii) its parameters $\pi_k$ (e.g., typed text, cursor coordinates); (iii) the screen state immediately before and after the action (keyframes or thumbnails); and (iv) short ASR transcripts spanning a 1-minute window before the action, the during span $[t_k^{\text{s}}, t_k^{\text{e}}]$, and a 1-minute window after. Conditioned on these inputs, the model outputs a concise rationale $r_k$ that states the intent, the local plan, and the expected state change

(grounded to visible UI). Additional prompt templates and representative inner-monologue examples are provided in Appendix F.

## 2.3 COMPUTER USE MODEL PRETRAINING

We demonstrate **VIDEOAGENTTREK**'s effectiveness by training an end-to-end computer-use agent with our video-driven data and a high-quality supervised finetuning set. On this strong finetuning basis, **VIDEOAGENTTREK** improves performance on online and offline agent evaluations.

### 2.3.1 AGENTIC DATA COLLECTION

**VideoAgentTrek Data.** We apply VIDEO2ACTION to the collected tutorial videos and convert them into agentic supervision. For each processed clip, we (i) run dense event detection to obtain typed, tightly bounded segments, (ii) infer action parameters with the action-identification recognizer, and (iii) generate a brief inner monologue for intent and expected state change. We then assemble the resulting steps $(I_k, r_k, a_k, \pi_k)$ into trajectories and serialize them for downstream training. In total, we processed 39,000 videos; each video produces on average 39 steps, yielding approximately 1.52 million ReAct steps overall, and about 26 billion training tokens. Detailed data statistics and examples will be provided in the Appendix J.

**Human demonstrations Data** We sample human-annotated trajectories from OpenCUA (Wang et al., 2025b) and AGUVIS (Xu et al., 2025b), harmonizing formats and labels into a single schema. The corpus spans Windows, macOS, and Android, contributing about 8B tokens to training.

**GUI Grounding Data.** We include a focused subset of GUI grounding pairs from the OSWorld-G dataset (Xie et al., 2025a) to strengthen pointer–target alignment and layout-aware perception. This contributes roughly 1B tokens to training.

### 2.3.2 TRAINING STRATEGY.

Automatically mined trajectories, while large-scale, inevitably contain residual noise. Motivated by prior findings that decoupling perception/grounding from policy learning improves robustness (Xu et al., 2025b; Wang et al., 2025b), we adopt a two-stage schedule that first stabilizes grounding on broad but imperfect supervision and then consolidates policy on a clean subset.

**Foundation** Qwen2.5-VL-7B (Bai et al., 2025) is a general vision-language model with superior vision understanding capability, but it is not sufficiently pretrained on computer-use tasks with an end-to-end success rate of 4.5% on OSWorld (Xie et al., 2024), which makes it a proper starting point (**base**) for evaluating the data generated by VIDEOAGENTTREK.

**Stage 1 training.** We train for one epoch over **26B** tokens drawn from the VideoAgentTrek trajectories, augmented with a small number of GUI grounding pairs. Trajectories are formatted as interleaved vision–text sequences: frames (or frame-equivalent images) appear inline with the step-wise textual outputs, preserving temporal order across the entire clip. Loss is masked to the textual portions only; images are conditioning context and are not predicted. Please refer to Appendix G for representative formatting examples and complete training configurations including hardware, batch sizes, and optimization details.

**Stage 2 training.** We continue training for **8B** tokens on a curated set of clean, human-annotated trajectories. Here we reformat the data into a chat template with user prompts and assistant responses that describe or execute the next action. We apply standard supervised finetuning with loss computed only on the assistant turns, leaving user turns as pure conditioning. Representative formatting examples and training details are provided in the Appendix G.

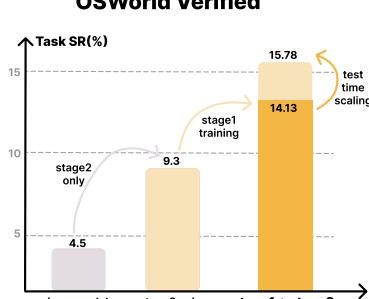

Figure 5: Experimental Results on OSWorld-Verified (Xie et al., 2025b) and AgentNetBench (Wang et al., 2025b). VideoAgentTrek demonstrates significant improvements over baseline models, with test-time scaling providing additional performance gains

# 3 EXPERIMENTS

## 3.1 COMPUTER USE AGENT PERFORMANCE

### 3.1.1 EXPERIMENT SETUP.

We evaluate the performance of our model on two computer-use agent benchmarks: OSWorld-Verified (Xie et al., 2025b; 2024) for online settings and AgentNetBench (Wang et al., 2025b) for offline settings. Further protocol, metrics, and computational details are provided in Appendix H.

1. **OSWorld-Verified.** OSWorld (Xie et al., 2024) is an online computer-use agent evaluation benchmark that includes 369 human-crafted Ubuntu desktop tasks. OSWorld-Verified (Xie et al., 2025b) is a more stable version, with updated evaluation scripts, environments, and clarified instructions, designed to measure CUA's task-solving capabilities in dynamic, real-world environments.

2. **AgentNetBench.** AgentNetBench is an offline benchmark is based on 100 representative tasks from the AgentNet dataset, covering a wide range of applications and websites on Windows and macOS. The tasks are manually refined and offer multiple valid action options for each step to reflect the variety of correct interactions.

### 3.1.2 MAIN RESULTS.

**Video pretraining enhances performance on offline benchmarks.** On AgentNetBench, incorporating VideoAgentTrek pretraining achieves a step success rate of **69.3%**, representing a 5.2 percentage point improvement over the SFT-only baseline (64.1%) and a substantial 30.8 percentage point gain over the base model (38.5%). This consistent improvement demonstrates that video pretraining effectively transfers knowledge to structured offline evaluation scenarios.

**Video pretraining delivers greater improvements on online benchmarks.** On OSWorld-Verified, our complete approach achieves a task success rate of 14.13%, demonstrating a 4.83 percentage point improvement (+52% relative) over SFT-only training (9.3%) and more than tripling the performance of the base model (4.5%). These gains are spread fairly evenly across different OSWorld-Verified application domains, rather than being concentrated in a single category.

**Video pretraining enables effective test-time scaling for computer-use agents.** In our setting, test-time scaling refers to increasing the allowed action-step budget during evaluation while keeping the trained agent fixed. On OSWorld-Verified, the model trained with Stage 1 and Stage 2 improves from 14.13% to 15.78% under test-time scaling, demonstrating that it can productively use additional interaction budget rather than simply taking more redundant steps. This benefit emerges specifically from video pretraining: models exposed to longer trajectories learn to plan over extended horizons, while the SFT-only baseline shows no improvement when evaluated under the same test-time scaling setup (see Section 4.1).

| Action | Preds | GT | Precision | Recall | F1 |
|--------|-------|-----|-----------|--------|-----|
| Click | 12,222 | 14,247 | 0.88 | 0.76 | 0.82 |
| Drag | 971 | 1,462 | 0.78 | 0.52 | 0.62 |
| Press | 177 | 842 | 0.40 | 0.08 | 0.14 |
| Scroll | 1,448 | 1,691 | 0.93 | 0.80 | 0.86 |
| Type | 1,480 | 2,040 | 0.89 | 0.64 | 0.75 |
| **Total** | **17,298** | **20,282** | **0.88** | **0.70** | **0.78** |

Table 1: Action-event detector evaluation: held-out test-set results by action type.

| Action type | Samples | Accuracy |
|-------------|---------|----------|
| Click | 324 | 0.713 |
| Drag | 22 | 0.366 |
| Press | 47 | 0.362 |
| Scroll | 34 | 0.735 |
| Type | 73 | 0.671 |
| **Overall** | **500** | **0.658** |

Table 2: Action parameterization evaluation: manual in-the-wild assessment.

## 3.2 VIDEO2ACTION PERFORMANCE

### 3.2.1 ACTION EVENT DETECTION

We assess VIDEO2ACTION with a two-part protocol: a held-out, annotated test set and an in-the-wild manual validation:

**Held-out test set.** We hold out 23 hours of screen-capture videos with 20,282 annotated GUI events. Each event is a tuple $(type, t^s, t^e)$. A prediction counts as a *hit* iff its type matches and its interval has any temporal overlap with a ground-truth event; unmatched predictions are false positives and unmatched ground truths are false negatives. We report per-type Precision/Recall/F1 and micro/macro aggregates.

**Manual validation (in-the-wild).** On 10 unseen YouTube tutorials, we apply the same overlap criterion and estimate recovery rates by human review to assess robustness outside the curated set.

**Results.** As the results shown in Table 1, Overall precision is high (0.88) with solid recall (0.71). Pointer-centric actions (click, scroll) are reliably localized; keystroke-only actions show lower recall/precision due to subtle visual evidence. In the manual study, the detector recovers ∼70% of actions under the same criterion, consistent with in-house results.

### 3.2.2 ACTION IDENTIFICATION

Evaluating action identification automatically is difficult because target-element boxes are unavailable. We therefore apply the identifier to in-the-wild videos and perform a blinded manual assessment. An action is judged *proper* if, when executed, it would plausibly produce the observed on-screen transition (for example, the clicked control changes state, typed text appears in the focused field, or the page scrolls). We evaluate 500 predictions sampled across action types.

**Results.** Annotators review pre/post frames and verify whether predicted parameters explain the observed changes, with disagreements resolved through second-pass review. Performance varies by action type: pointer-based actions (click, scroll) achieve highest accuracy, typing shows moderate accuracy despite OCR noise, while drag/press actions struggle with subtle visual cues. Despite these challenges, the predicted parameters are accurate enough for trajectory construction and downstream training; detailed counts and validation rates appear in Table 2.

## 4 ANALYSIS

### 4.1 EFFECTIVENESS OF DATA SCALING

To assess the impact of Stage-1 data scale, we train models using 0%, 50%, and 100% of the video tokens, then apply identical Stage-2 SFT to each variant. With increasing tokens, performance scales consistently across both benchmarks. Agent-NetBench step success rates increase from 64.1% to 68.1% and 69.3%, while OSWorld-Verified task SR@50 grow from 9.3% to 13.3% and 15.7% (Figure 6). These findings establish a clear

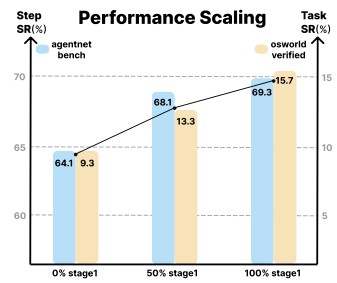

Figure 6: Performance Scaling

relationship between pretraining data size and computer-use
agent performance, demonstrating the benefits of scaling video
pretraining data.

## 4.2 Improving Long Horizon planning

VIDEOAGENTTREK provides substantially longer trajectories than previous CUA corpora. As
illustrated in Figure 13, 42.1% of trajectories exceed 20 steps, while 14.5% contain 50 or more
steps, yielding an average trajectory length of 39.25 steps. Cross-dataset comparisons (Table 12)
reveal that this average substantially exceeds those of established benchmarks, demonstrating that
VIDEOAGENTTREKcorpus emphasizes supervision of complex, multi-step workflows rather than
brief, single-interaction sequences.

The benefits of long-horizon supervision become evident when evaluating planning capabilities under
varying step budgets. On OSWorld-Verified, we observe a striking difference in how models respond
to increased action budgets. The Stage2-only model shows no performance improvement when the
step budget expands from 20 to 50 steps, remaining flat at 9.3% task success, indicating it cannot
effectively plan beyond its training horizon or recover from early mistakes. In contrast, after Stage-1
pretraining on VideoAgentTrek's long video trajectories, the agent demonstrates true test-time scaling:
task success rate increases from 14.13% at 20 steps to 15.78% at 50 steps, a +1.65 point absolute
improvement (+11.7% relative gain, Figure 5). This differential reveals that exposure to extended
video demonstrations during pretraining teaches the model to decompose complex tasks into subgoals,
persist through intermediate failures, and leverage additional computational budget for exploration
and error correction, capabilities that supervised fine-tuning on shorter trajectories fails to instill.

## 5 Related Work

**Generating agent trajectories.** Computer-use trajectories have been obtained through human
annotation, programmatic synthesis in instrumented environments, and web-scale mining of pub-
lic resources. Human annotation, often aided by instrumentation to log pointer coordinates and
keystrokes, yields precise labels but is costly and narrow in coverage (Qin et al., 2025; Wang et al.,
2025a;b). Programmatic synthesis inside headless browsers or scripted desktop flows can generate
large volumes with exact parameters, yet coverage is constrained by simulator APIs and may diverge
from real-world UI variability (Su et al., 2025; Sun et al., 2025). Web-scale mining taps tutorials,
RPA logs, and screen recordings to obtain diverse trajectories, but typically lacks precise temporal
boundaries or action parameters (Xu et al., 2025a; Jang et al., 2025).

**Precise Event Grounding in Video.** Temporal grounding approaches such as temporal action local-
ization, moment retrieval, keyframe detection, and dense video captioning seek to determine when
events take place and provide corresponding descriptions (Lin et al., 2019; Zhuang et al., 2025; Wasim
et al., 2024). Meanwhile, recent multimodal systems (e.g., Qwen2.5-VL (Bai et al., 2025), Gemini
2.5 Pro (Comanici et al., 2025)) have advanced the field by enabling more detailed spatiotemporal
understanding and long-horizon video reasoning. Nonetheless, most general-purpose grounding
frameworks focus primarily on semantic interpretation, rather than achieving the millisecond-level
precision and parameter extraction required to faithfully reconstruct GUI interactions.

**Learning from unlabeled video to act in environments.** VPT demonstrated that large-scale
*unlabeled* videos can be converted into effective training signals (e.g., via inverse-dynamics auto-
labeling followed by behavior cloning), substantially improving an agent's ability to act (Baker et al.,
2022).Building on this idea, subsequent work leverages internet-scale human videos to distill human
policy priors that transfer to interactive environments, including learning action-centric latent spaces
without action labels (Ye et al., 2025) and scaling to humanoid control (Mao et al., 2024).

## 6 Conclusion

We presented VideoAgentTrek, a scalable pipeline that transforms publicly available screen recordings
into structured supervision for computer-use agents without manual annotation. By developing an
inverse dynamics module that accurately detects GUI events and extracts action parameters from raw
video, we demonstrate that the implicit supervision in tutorial videos can be effectively harvested at

scale. Our experiments on 39,000 YouTube videos yielded 1.52 million interaction steps, enabling continued pretraining that improved task success rates by 70% on OSWorld-Verified (9.3% to 15.8%) and increased step accuracy on AgentNetBench from 64.1% to 69.3%. These results establish that unlabeled internet videos, when processed through learned inverse dynamics, provide a viable and cost-effective alternative to expensive manual annotation for training robust computer-use agents. The open-source release of our ScreenFilter and Video2Action tools enables the community to leverage this abundant resource for advancing GUI automation research.

## 7 ACKNOWLEDGEMENTS

The authors of this paper were supported by the ECS (27212023) and Areas of Excellence Scheme (AoE/E-601/24-N) from RGC of Hong Kong.

**Reproducibility Statement**   We have made extensive efforts to ensure the reproducibility of our work. Our pipeline consists of three main components, each with detailed specifications provided in the appendix: (1) **ScreenFilter**: Training details including the 15,000 synthetic training images, model architecture, and performance metrics (89.58% F1 score) are provided in Appendix C. The model processes videos at 1–2 fps with cursor detection thresholds and temporal smoothing parameters fully specified. (2) **Video2Action**: The inverse dynamics module training on 154 hours of OpenCUA data is detailed in Appendix D, including the complete action taxonomy, training configuration, and evaluation metrics. Action parametrization methods are described in Appendix E. (3) **Agent Training**: Complete training configurations for both Stage-1 continued pretraining and Stage-2 fine-tuning are provided in Appendix G, including learning curves, Megatron-LM framework settings, and tensor/pipeline parallelism configurations. The VideoAgentTrek dataset construction process, including YouTube video selection criteria (Appendix B), data statistics, and quality validation results are documented in Appendix J. Upon publication, we will also release a dataset manifest containing YouTube Video IDs for all processed videos and full action metadata for all steps (action types, parameters, inner thoughts, timestamps), together with a reconstruction script that can automatically download and preprocess the corpus from this manifest. In addition, we will provide a small Creative Commons–licensed subset whose trajectories, screenshots, and metadata can be shared directly, enabling exact replication of Stage-1 pretraining on a miniature scale. Evaluation protocols for OSWorld-Verified and AgentNetBench benchmarks are specified in Appendix H. We will release the ScreenFilter and Video2Action tools as open-source software with pretrained checkpoints upon publication. The processed VideoAgentTrek dataset will be made available subject to YouTube's terms of service. All LLM usage for data auditing and annotation generation has been fully disclosed with prompts provided (Appendix F).

**Ethics Statement**   We carefully considered the ethical implications of our work in developing VideoAgentTrek. **Data Collection and Fair Use**: We accessed YouTube videos through public APIs in compliance with YouTube's Terms of Service, using content for non-commercial research purposes under fair use principles. Videos were processed in a streaming manner without permanent storage, and we respect content creators' rights by not redistributing raw video data. **Privacy Protection**: Our pipeline is designed to focus exclusively on GUI interactions and application interfaces, not personal content. The ScreenFilter module specifically detects cursor movements and interface elements while avoiding processing of personal information such as emails, documents, or user credentials visible in recordings. For training, we only use text that is already visible in public tutorial videos and do not attempt to infer hidden content (for example, masked passwords), and for any released manifests or textual metadata we apply conservative PII filters (for example, removing email-like strings or obvious credentials) so that our artifacts do not expose more sensitive information than the original public videos. **Potential Misuse and Dual-Use Considerations**: We acknowledge that automation technologies can be misused for unauthorized scraping, bot creation, or circumventing application restrictions. To mitigate risks, we emphasize that our tools are intended for research and accessibility applications, and we encourage users to respect application terms of service, obtain appropriate permissions for automation, and consider the impact on human workers whose tasks might be automated. The released models include documentation on responsible use guidelines. **Transparency and Reproducibility**: We provide full disclosure of LLM usage in data processing. Our open-source release enables scrutiny of methods while our evaluation includes manual validation of extracted actions to ensure quality. We believe the benefits of advancing computer-use agents for accessibility, automation, and human-computer interaction research outweigh the risks when appropriate safeguards are implemented.

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

## A    LLM USAGE DISCLOSURE

**Scope.** We used large language models in three ways: (i) data auditing, (ii) dataset annotation, and (iii) drafting/formatting text. All experimental results, figures, and quantitative claims were produced from our own training/evaluation runs and were reviewed by the authors.

**Data auditing (metadata classification).** A lightweight GPT‑4.1 pass over video titles and short descriptions was used to assign coarse content-form labels (tutorial/background/tech talk/other) and OSWorld-style domain tags. These labels supported corpus auditing and analysis and did not directly determine benchmark results.

**Dataset annotation (inner monologue).** For the inner-monologue component of ReAct trajectories, we prompted an LLM (GPT‑5 Medium) with structured evidence (action type/content, pre/post keyframes, and short ASR snippets) to generate step-level rationales. Prompts and examples are provided in Appendix F. Outputs were constrained by explicit rules (e.g., no coordinates, first-person reasoning) and were filtered by automatic checks (consistency with $(a_k, \pi_k)$, basic PII redaction) followed by human spot-review.

**Manuscript drafting and LaTeX.** An LLM assistant was used to help reorganize and polish prose, and to format LaTeX snippets (e.g., converting paragraphs to concise task definitions, producing wrapfig/wraptable layouts). All technical content, equations, numbers, and citations were written or verified by the authors; no passages were accepted verbatim without human editing.

**Coding and experiment orchestration.** Training, data processing, and evaluation scripts were authored and run by the authors. LLMs were not used to autonomously orchestrate experiments or to fabricate results.

**Human validation.** Authors reviewed prompts, sampled model-generated annotations for fidelity to the visual/audio evidence, and checked all reported metrics and tables against logs. If an LLM was used only for grammar or wording, its suggestions were edited and verified by the authors.

**No undisclosed assistance.** Beyond the uses above, no additional LLM assistance was employed.

## B    YOUTUBE VIDEO QUALITY STANDARDS

To ensure consistency and usability in selecting high-quality instructional videos from YouTube for research purposes, the following standards must be met:

1. **Minimal Overlays.** If overlays, such as face cams or titles, are present, they must occupy no more than $\frac{1}{10}$ of the screen area to avoid obstructing the primary content.

2. **Primary Focus on Screen Recording.** The video should predominantly feature clean screen recordings. Brief transitions to other scenes, such as PowerPoint slides or face capture, are permissible but should be limited to introductory or concluding segments.

3. **Screen Recording Method.** The video must consist of direct screen recordings rather than footage captured by an external camera.

4. **Language Requirement.** The video must be in English to facilitate monolingual captioning in subsequent processing steps.

5. **Stable Visual Presentation.** Frequent zooming in or out should be avoided. The entire screen or application window must be visible for the majority of the video duration.

6. **Caption Availability.** The video must include captions, indicated by the availability of the closed caption (CC) icon in the bottom right corner of the player. Captions may be auto-generated or manually annotated.

7. **Orientation.** The video must be recorded in a horizontal format, as vertical videos often fail to capture complete desktop screens, limiting their utility.

8. **Recency.** Videos must be no older than five years to ensure that the user interfaces depicted remain relevant and applicable.

These criteria ensure that selected videos are suitable for detailed analysis and processing in research contexts.

Table 3: Cursor Type Distribution in Training Data

| macOS (39 types) | | | | Windows (19 types) | | |
|---|---|---|---|---|---|---|
| **Icon** | **Type** | **%** | | **Icon** | **Type** | **%** |
| | Default | 25.0 | | | Aero Move | 2.0 |
| | Hand Point | 20.0 | | | Move (IL) | 2.0 |
| | Text | 15.0 | | | Move (L) | 2.0 |
| | Cell | 5.0 | | | Size 1 (IL) | 1.0 |
| | Cross | 5.0 | | | Size 1 (L) | 1.0 |
| | Hand Grab | 5.0 | | | Size 2 (IL) | 1.0 |
| | Hand Open | 5.0 | | | Size 2 (L) | 1.0 |
| | Move | 5.0 | | | Size 3 (IL) | 1.0 |
| *+ 31 others* | | 15.0 | | | Size 3 (L) | 1.0 |
| | | | | | Size 4 (IL) | 1.0 |
| | | | | | Size 4 (L) | 1.0 |
| | | | | *+ 8 default & others* | | 86.0 |

## C  SCREENFILTER DETAILS

SCREENFILTER is trained on 15,000 synthetic images generated by compositing cursor sprites onto GUI screenshots from the GUIEnv (Chen et al., 2025) dataset. To enhance its generalization across different platforms, we incorporate various cursor patterns from both Windows and macOS. Specifically, we collected 39 common cursor types from macOS and 19 common cursor types from Windows. Detailed cursor specifications and their distribution in the training data are provided in Table 3. On the held-out test set, SCREENFILTER achieves an F1 score of 89.58%, with 90.64% precision and 88.54% recall, demonstrating its effectiveness in accurately separating computer-use content from unrelated material.

For video processing, SCREENFILTER operates at 1-2 frames per second to balance both accuracy and efficiency. The model retains segments where at least 80% of the frames contain a cursor for a minimum of 6 seconds, with a 2-second temporal smoothing gap to merge frames. This design allows SCREENFILTER to process approximately 840 hours of video per GPU-day, facilitating large-scale filtering.

## D  DENSE EVENT DETECTION

Our dense detector is trained on **154 hours** of screen-capture video paired with raw interaction logs from *OpenCUA* (Wang et al., 2025b). The logs contain complete demonstrations with precisely timestamped GUI interactions (mouse and keyboard). We convert these logs into prompt-free temporal grounding supervision by mapping low-level events to our action taxonomy, merging short consecutive micro-events into typed spans with start and end timestamps, and discarding segments without actionable GUI operations.

For training-set preparation, we downsample videos to 4 fps, segment them into non-overlapping 10 s clips, and align the interaction logs within each clip to obtain typed spans with start and end timestamps. We adopt a temporal patch size of 2 frames for modeling efficiency. Label names are normalized to our action taxonomy. We visualize the data sample in Figure 7

**Model training.** We perform full-parameter supervised fine-tuning of Qwen2.5-VL-7B-Instruct. The training configuration and loss curve are shown side-by-side in Figure 8.

**Evaluation protocol.** We evaluate dense event detection at the segment level on a 0.5 s temporal grid and use different overlap criteria for short and long actions. Concretely, we consider the following action types: `click`, `key`, `write (type)`, `scroll`, `moveTo`, `dragTo`, `doubleClick`, `rightClick`, `hscroll`, `hotkey`, `tripleClick`, `middleClick`. For short, instantaneous interactions such as `click`, `key`, `hotkey`, `scroll`, and `doubleClick`,

| Action type | Count |
|---|---|
| click | 410,101 |
| key | 138,660 |
| write | 80,749 |
| scroll | 46,597 |
| moveTo | 32,840 |
| dragTo | 32,840 |
| doubleClick | 14,241 |
| rightClick | 7,451 |
| hscroll | 3,411 |
| hotkey | 2,570 |
| tripleClick | 2,428 |
| middleClick | 57 |
| **Total** | **771,945** |

Table 4: Event distribution in the dense event detection training data (154 hours).

| Resolution | Clips |
|---|---|
| 1280×720 | 9,854 |
| 1212×758 | 5,950 |
| 1276×718 | 1,865 |
| 1192×772 | 1,679 |
| 1172×782 | 852 |
| 1188×772 | 604 |
| 2560×1440 | 574 |
| 1188×774 | 471 |
| 1192×770 | 417 |
| 1468×956 | 182 |
| other | 806 |
| **Total** | **23,254** |

Table 5: Resolution distribution of dense event detection training clips.

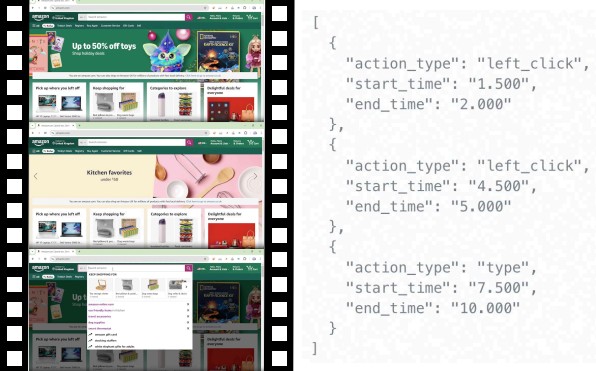

Figure 7: Example training sample for the dense event detector.

the raw interaction duration is very brief and the exact delay between input and visible UI change is ambiguous. We therefore represent these as short segments aligned to the 0.5 s grid and count a prediction as correct if the predicted segment has any non-zero temporal overlap with the ground-truth segment. For longer actions, especially text-input events (`write`/`type`), we require that the predicted segment cover at least 80% of the ground-truth duration, i.e., $|\text{intersection}|/|\text{GT}| \geq 0.8$, so that a hit reflects capturing most of the actual typing episode. We will release the evaluation scripts so that future work can reproduce and compare event grounding performance under the same temporal-overlap criteria.

## E  ACTION IDENTIFICATION

**Training data.**

Our dense detector is trained on 512,000 screen-capture clips paired with raw interaction logs from *OpenCUA* (Wang et al., 2025b). To preprocess action segments, we adopt a *dynamic frame-rate* policy that caps frames per clip at 20 while preserving short, fast interactions. For a segment of duration $\Delta t$ (seconds), we set

$$f = \min\{30, \ \max\{4, \ \lfloor 20/\Delta t \rfloor\}\},$$

then sample frames uniformly within $[t_k^s, t_k^e]$. This yields, for example, $f=30$ for brief clicks/scrolls ($\Delta t \approx 0.5$ s, $\approx 15$ frames), $f \approx 20$ for $\Delta t \approx 1.0$ s (20 frames), and $f=4$ for extended typing segments ($\Delta t \approx 5$ s, 20 frames). We visualize the data sample in Figure

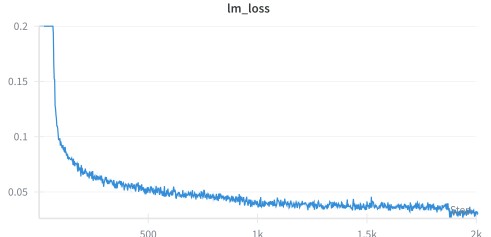

| | |
|---|---|
| Framework | Megatron-LM |
| Hardware | $32\times$ H100 GPUs |
| Tensor parallelism | TP = 4 |
| Pipeline parallelism | PP = 1 |
| Global batch size | 256 |
| Training iterations | 2000 |
| LR decay iterations | 2000 |
| Wall-clock time | $\sim$15 h |

Figure 8: Dense event detector: training loss (left) and training configuration (right).

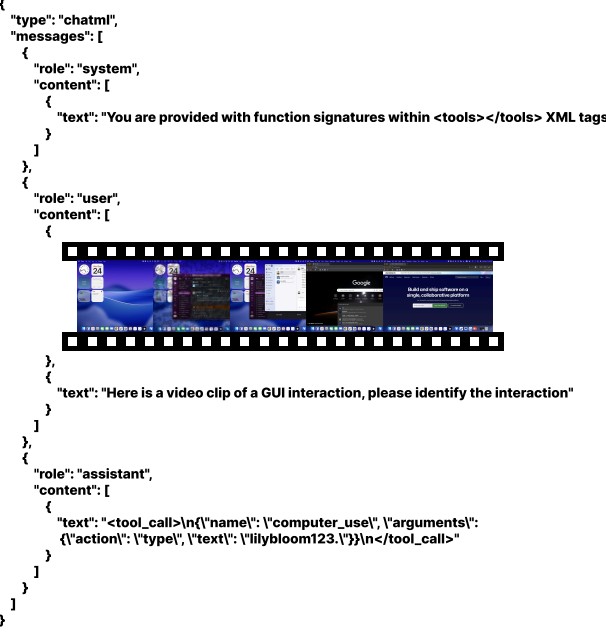

Figure 9: Example training sample for the action parametrization model.

**Stage-1 / Stage-2 consistency handling.** For each detected event, Stage 1 outputs an action type and temporal span $(a_k^{\text{det}}, t_k^{\text{s}}, t_k^{\text{e}})$, while the Stage 2 parametrization model, given the corresponding clip, predicts a full tool call with its own action type $a_k^{\text{para}}$ and parameters (for example, coordinates or text). In post-processing, we compare these two: if $a_k^{\text{para}}$ matches $a_k^{\text{det}}$, we directly take the Stage 2 prediction as the final action; if they disagree, we treat the case as potentially noisy, re-run Stage 2 with the type fixed to $a_k^{\text{det}}$ to obtain an alternative candidate, and then ask `gpt-5-medium`, given the pre-/post-action frames and both candidates, which action better explains the observed state transition. The chosen candidate is used as the final label, allowing Stage 2 to benefit from joint type+parameter prediction while using Stage 1 as a stabilizing prior in ambiguous cases.

# F INNER MONOLOGUE GENERATION

**Prompt Content.**

Inputs:
Action type: $a_k$    Parameters: $\pi_k$
Before/after keyframes: $I_k^{\text{pre}}$, $I_k^{\text{post}}$
ASR windows: $[-60\text{s}, 0], [t_k^{\text{s}}, t_k^{\text{e}}], [0, 60\text{s}]$

**Instruction (to model):**
You are generating inner-monologue annotations for a dataset of GUI agent trajectories built from in-the-wild screen recordings.

**End-to-end setting.**
- Source: real GUI screen recordings from the wild.
- Extraction: each GUI interaction (an action) is automatically detected from video/audio.
- For every detected action, you receive three kinds of evidence:
  - *Action details*: {action_type} and {action_content}.
    action_content may contain: coordinates (absolute or normalized) and/or a bbox; typed text; pressed keys; scroll amount/direction; drag start/end; and similar specifics.
  - *Keyframes*: a start screenshot and, if available, an end screenshot right after the action executes.
  - *Surrounding transcripts*: short snippets of narration or speech immediately before, during, and after the action.
  - *Action validation (optional)*: a brief validator description summarizing what occurred.

**Your task.** For each action, output **exactly one** JSON object with two fields: *action_description* and *thought*.

**Field definitions (strict).**
- **action_description**: a concise natural-language description of *what* I do in the UI at this step. Name the target UI element if inferable (button, menu, tab, field); otherwise describe by role/label/relative position. Mention the immediate visible outcome only if it is clearly observable. **Forbidden**: raw coordinates, code, function/method names, automation tokens, key–value argument lists.
- **thought**: my first-person inner monologue (4–8 sentences) as the demonstrator (use "I", "me", "my"). Provide substantive reasoning. Include: (a) what I aim to accomplish and why now; (b) how the speech context informs my intent (weave naturally); (c) a brief summary of what likely changes from start to end if both frames exist; (d) a short breakdown of the atomic actions in this step (e.g., type + press) and why each is needed; (e) what I expect to verify or do next. Prefer present tense when natural.

**General rules.**
- The thought must be in first person; never switch to third person.
- Evidence priority: prefer visual evidence from start/end keyframes; treat speech as a weak hint for *why*. If they conflict, prefer visuals.
- Weave evidence naturally without naming "transcripts" or "frames."
- For coordinate-based actions, a red hollow circle may mark the interaction point; **do not mention** the marker, describe the target element instead.
- If only a start keyframe is available, focus on intent; if an end keyframe exists, you may include the immediate visible result.
- When a step bundles multiple atomic actions, reason across them as one coherent operation.
- Keep *action_description* concise; let *thought* carry the details; avoid hedging and boilerplate.
- **Output format**: exactly one valid JSON object with only *action_description* and *thought*; no extra keys or commentary.

**Output:**
$r_k$: inner-monologue JSON with fields *action_description* and *thought*.

**Qualitative analysis.** To further sanity-check the reliability of the generated inner monologue, we conducted a small-scale human review on a fixed validation subset of 100 step-wise samples spanning diverse scenarios (for example, macOS Mail, Windows Calculator, macOS Preview, Chrome browsing). For each action step, we examined whether (i) the *action_description* matches the underlying tool call type and target element, (ii) the *thought* correctly captures the local intent of

the step, and (iii) the expressed intent is reasonably consistent with how the narrator conveys it in the transcript. On this subset, 71% of steps satisfied all three criteria. Among the remaining 29%, around 7% were due to incorrect action recognition, 12% corresponded to inherently noisy or exploratory operations whose scene transitions were not meaningfully goal-directed, and 10% came from non–goal-oriented segments with unclear narrator intent, where the inner monologue showed mild hallucination but remained acceptable for pretraining. Overall, 81% of the reviewed steps were judged qualified for Stage-1 pretraining.

**Sample inner monologue.** Figure 10 shows a representative example from our 20-trajectory validation subset: given a pre-action frame, the automatically detected action, and the post-action frame, the model generates a step-wise inner monologue that explains the user's intent (for example, opening a specific photo event in iPhoto), the immediate UI operation, and the expected outcome.

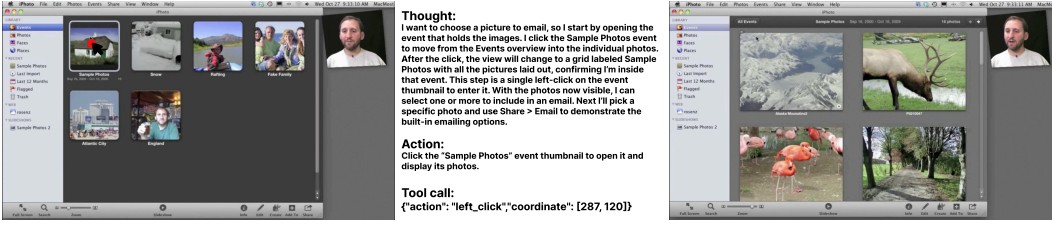

Figure 10: Example of generated inner monologue for a single GUI action. Left: pre-action frame with the target element highlighted. Middle: generated thought and natural-language action description, together with the structured tool call. Right: post-action frame showing the resulting UI state.

## G    COMPUTER USE AGENT TRAINING

**Training Data.** we visualize the training data samples in stage-1 and stage-2 training in Figure 11.

**Stage-1 training.** We perform full-parameter continue-pretraining Qwen2.5-VL-7B-Instruct. The training configuration and loss curve are shown side-by-side in Figure 12.

**Effect of Stage-1 pseudo-label noise on grounding.** Stage-1 video continued pretraining is designed to inject long-horizon computer-use knowledge, but it is trained on automatically extracted pseudo-trajectories whose parameters are not perfectly clean. To quantify how such noise manifests in low-level skills, we evaluate all model variants on OSWorld-G, a dedicated grounding benchmark that directly probes the ability to localize and select the correct on-screen target. Table 6 summarizes the results alongside OSWorld-Verified and AgentNetBench.

| Model variant | OSWorld-G | OSWorld-Ver. | AgentNetBench | WindowsAA |
|---|---|---|---|---|
| Qwen2.5-VL-7B (base) | 31.40 | 4.50 | 38.5 | 7.80 |
| + Stage-2 SFT only | 31.56 | 9.30 | 64.1 | 15.38 |
| + Stage-1 CPT only | 26.24 | – | – | – |
| Stage-1 CPT + Stage-2 SFT | 30.50 | 15.78 | 69.3 | 17.39 |

Table 6: Effect of Stage-1 video continued pretraining and Stage-2 supervised fine-tuning on grounding (OSWorld-G) and end-to-end performance across benchmarks (OSWorld-Verified, AgentNet-Bench, Windows Agent Arena with 50-step budget).

**Stage-2 training.** We continue a full-parameter supervised fine-tuning on the stage-1-trained checkpoint. The training configuration and loss curve are shown side-by-side in Figure 13.

## H    COMPUTER USE AGENT EVALUATION

**Evaluation Setting.** We follow the OSWorld-Verified protocol: the agent interacts with a live desktop given a natural-language instruction and the full history of prior states and actions. At each step, the policy conditions on the instruction and a bounded visual context of up to five recent screenshots

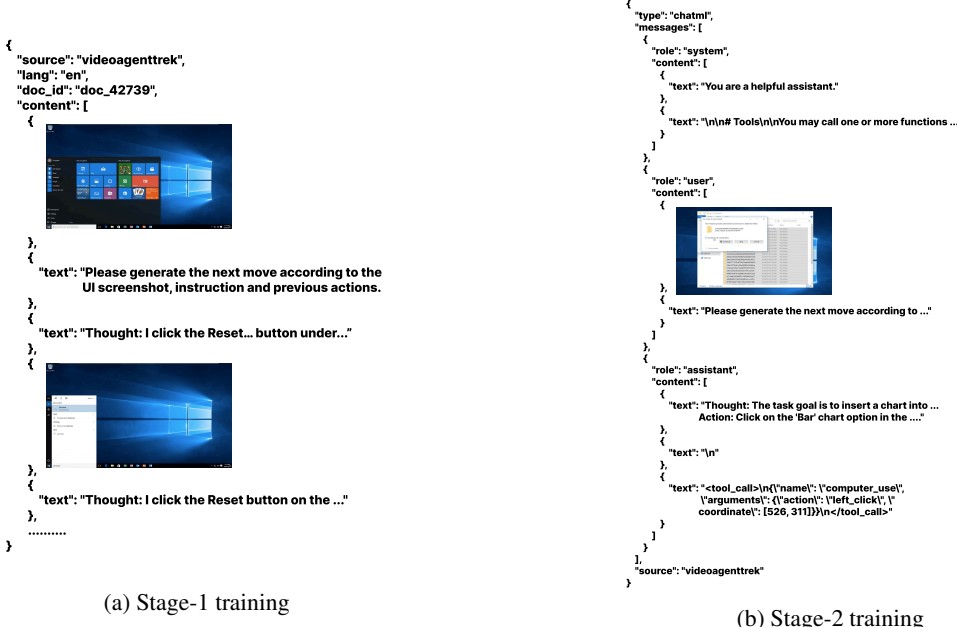

(a) Stage-1 training

(b) Stage-2 training

Figure 11: Computer Use Agent Training Data (a) Stage-1 training, (b) Stage-2 training.

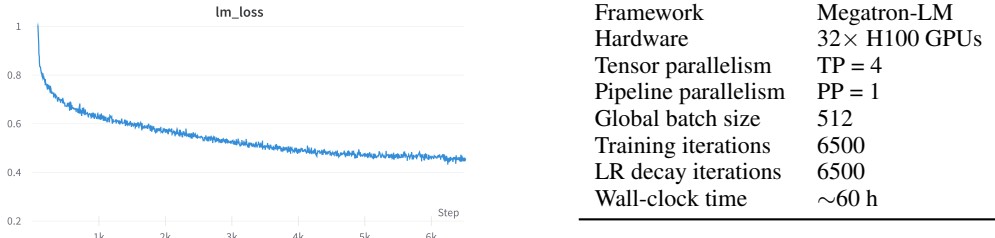

| Framework | Megatron-LM |
|---|---|
| Hardware | 32× H100 GPUs |
| Tensor parallelism | TP = 4 |
| Pipeline parallelism | PP = 1 |
| Global batch size | 512 |
| Training iterations | 6500 |
| LR decay iterations | 6500 |
| Wall-clock time | ∼60 h |

Figure 12: CUA Stage-1 Training: training loss (left) and training configuration (right).

(FIFO window) together with the action/rationale history, then emits the next action. For the 20-step budget, we conduct three independent runs per model and report the average Task SR. For the 50-step budget, we perform a single run. All models use identical inference settings and action executors; no manual interventions are allowed during evaluation.

**AgentNet Bench summary.** Overall step SR rises from 0.385 (base) to 0.641 with SFT-only and to **0.693** with Stage 1 + Stage 2. These trends suggest that video pretraining notably strengthens grounding and multi-action control, especially for less frequent or harder motor primitives.

**OSWorld-Verified summary.** Table 7 reports task success across turns and step budgets. With SFT-only (*stage2 only*), Task SR hovers around 9.1–9.4% at 20 steps and shows no improvement at 50 steps (9.27%), indicating limited ability to leverage longer budgets. Adding VideoAgentTrek pretraining (*stage1 + stage2*) raises Task SR to 13.6–14.7% at 20 steps and further to **15.78%** at 50 steps. Per-domain counts improve most for *chrome/46* (up to **15** solved) and *workflow/92* (up to **8** solved), with steady gains in *os/24* and authoring apps (writer, impress). Across three 20-step runs, variance is modest, suggesting stable benefits from Stage 1. Overall, the results show that large-scale video pretraining yields higher step quality and makes the agent budget-sensitive—able to convert extra steps into additional task completions.

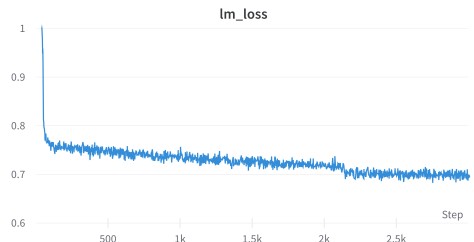

| Framework | Megatron-LM |
|---|---|
| Hardware | 64× H100 GPUs |
| Tensor parallelism | TP = 4 |
| Pipeline parallelism | PP = 1 |
| Global batch size | 512 |
| Training iterations | 3000 |
| LR decay iterations | 3000 |
| Wall-clock time | ∼16 h |

Figure 13: CUA Stage-2 Training: training loss (left) and training configuration (right).

| Model | Eval | Task SR (%) | calc | chrome | gimp | vscode | writer | tbird | os | impress | workflow | vlc |
|---|---|---|---|---|---|---|---|---|---|---|---|---|
| stage2 only | turn1 (20) | 9.42 | 2.2 | 17.4 | 11.5 | 26.1 | 8.7 | 20.0 | 8.3 | 6.4 | 5.4 | 5.9 |
| | turn2 (20) | 9.13 | 4.3 | 17.4 | 3.8 | 26.1 | 8.7 | 13.3 | 8.3 | 6.4 | 5.4 | 11.8 |
| | turn3 (20) | 9.42 | 2.2 | 17.4 | 7.7 | 26.1 | 8.7 | 13.3 | 8.3 | 8.5 | 5.4 | 11.8 |
| | turn4 (50) | 9.27 | 2.2 | 26.1 | 7.7 | 13.0 | 8.7 | 13.3 | 8.3 | 8.5 | 4.3 | 5.9 |
| stage1 + stage2 | turn1 (20) | 14.68 | 4.3 | 28.3 | 7.7 | 26.1 | 21.7 | 40.0 | 16.7 | 12.8 | 7.6 | 11.8 |
| | turn2 (20) | 13.57 | 4.3 | 28.3 | 7.7 | 21.7 | 21.7 | 40.0 | 12.5 | 12.8 | 5.4 | 11.8 |
| | turn3 (20) | 14.13 | 4.3 | 26.1 | 7.7 | 30.4 | 30.4 | 40.0 | 12.5 | 10.6 | 5.4 | 11.8 |
| | turn4 (50) | **15.78** | 2.2 | **32.6** | 7.7 | 26.1 | 26.1 | 40.0 | 16.7 | 12.8 | **8.7** | 17.6 |

Table 7: OSWorld-Verified full results. Cells indicate per-application success rates (percent of solved tasks in each bucket).

# I    VIDEOAGENTTREK DATA ANALYSIS

**Resolution and scale.** We downloaded 55,603 screen-capture videos (about 10,000 hours) from 50+ channels. The corpus is predominantly clear: 97% are 720p or higher (Table 9). Most videos are minutes long, providing sustained interactions suitable for dense detection and action identification.

**Title/description-based content classification.** To quickly audit topical relevance at scale, we apply a lightweight classifier to each video's title and brief description.

- **Labels.**
  - **A_tutorial**: hands-on screen tutorials.
    * *Include*: step-by-step demonstrations, cursor-driven walkthroughs, "how to . . ." tasks; frequent UI focus changes; imperative phrasing in titles ("Create. . .", "Install. . .", "Fix. . .").
    * *Exclude*: talk-style narrations with few concrete on-screen actions; marketing teasers without real steps.
    * *Signals*: verbs tied to UI operations (open, click, type), timestamps/chapters per step, tool/app names plus action verbs.
  - **B_background**: expository background with incidental screen use.
    * *Include*: market share reports, product overviews, concept explainers where the desktop appears only as a backdrop.
    * *Exclude*: segments that actually show multi-step operations (move to A_tutorial).
    * *Signals*: nouns like "overview, history, comparison, review," charts/stats in title/description, little or no cursor interaction.
  - **C_tech_talk**: talks or presentations with slides.
    * *Include*: conference talks, webinars, lectures; slide navigation with limited live demos.
    * *Exclude*: talks that transition into substantial live step-by-step demos (then split or relabel A_tutorial).
    * *Signals*: "keynote, webinar, seminar, lecture," speaker names/affiliations, slide thumbnails.
  - **D_unrelated**: off-topic for computer-use learning.
    * *Include*: content where a screen appears but no actionable computer-use task is taught (e.g., pure entertainment, face-cam only).

| Model | Step SR | click | write | press | scroll | moveTo | dragTo | hotkey | dbClick | rClick | terminate |
|---|---|---|---|---|---|---|---|---|---|---|---|
| base | 0.385 | 0.402 | 0.605 | 0.286 | 0.615 | 0.189 | 0.000 | 0.250 | 0.000 | 0.000 | 0.188 |
| stage2 only | 0.641 | 0.671 | 0.719 | — | 0.500 | 0.300 | 0.145 | 0.484 | 0.526 | 0.214 | 0.588 |
| stage1 + 2 | **0.693** | **0.767** | **0.733** | **0.441** | 0.600 | **0.502** | **0.264** | **0.562** | **0.650** | **0.417** | 0.237 |

Table 8: AgentNet Bench: step success rate (overall and per action type). "—" indicates the metric was not applicable/recorded.

| Resolution bucket | Count |
|---|---|
| High (1080p+) | 2,322 |
| Standard (720p–1080p) | 49,589 |
| Low (<720p) | 1,464 |

Table 9: Resolution distribution of downloaded videos.

* *Exclude*: any clip with consistent stepwise UI operations (move to A_tutorial).
* *Signals*: lifestyle/vlog tags, gameplay without UI instruction, no app/task keywords.

- **Procedure.** Single-pass GPT-4.1 classification with a short instruction to choose exactly one of the four labels given the title and short description; no transcript or frames are used. We use the result only for corpus auditing, tag mining, and optional down-weighting in later filters, not as a hard accept/reject gate.

- **Limitations.** Metadata-only classification can mislabel borderline cases (e.g., talks that include substantial demos). Final training sets are still screened by cursor gating, license/PII checks, and downstream detectors.

**Distribution.** Class counts and shares are shown in Table 10. The majority are tutorials, indicating strong alignment with our target use case.

| Label | Count | Share |
|---|---|---|
| A_tutorial | 38,700 | 69.6% |
| B_background | 12,900 | 23.2% |
| C_tech_talk | 2,391 | 4.3% |
| D_unrelated | 1,612 | 2.9% |
| **Total** | **55,603** | **100%** |

Table 10: distribution from title/description classification.

| Action type | Count | Share (%) |
|---|---|---|
| left_click | 1,037,617 | 67.1 |
| type | 214,816 | 13.9 |
| key | 145,860 | 9.4 |
| scroll | 111,203 | 7.2 |
| right_click | 24,111 | 1.6 |
| double_click | 11,848 | 0.8 |
| mouse_move | 8,441 | 0.1 |
| drag | 6,372 | 0.1 |
| hscroll | 196 | 0.0 |
| **Total** | **1,547,092** | **100.0** |

Table 11: Action distribution in the VideoAgent-Trek agentic dataset.

**Action distribution.** Table 11 summarizes action counts in the VideoAgentTrek agentic data.

**Cross-dataset comparison.** We summarize reported average step counts (and task counts when available) for common CUA datasets and include our corpus for context.

**Benchmark contamination analysis.** Our pretraining corpus is mined from long-form tutorial videos, which differ conceptually from the goal-conditioned tasks in OSWorld and AgentNetBench: tutorials typically demonstrate a range of operations on ad-hoc content (for example, showing several Excel functions on randomly created tables) rather than executing a single explicit instruction toward a precise target state. To visualize this gap, we encode all tutorial video titles and all OSWorld task instructions into a shared text-embedding space and project them into 2D (Figure 14). OSWorld instructions form several relatively compact clusters that cover only a small portion of the overall tutorial distribution, while the video titles are widely dispersed across many unrelated domains and workflows. We also verify that none of the titles contain the strings "OSWorld" or "AgentNetBench",

| Dataset | Tasks | Avg. Step |
|---|---|---|
| AndroidControl (Li et al., 2024) | 15,283 | 5.5 |
| AMEX (Chai et al., 2025) | 2,991 | 11.9 |
| AitW (Rawles et al., 2023) | 2,346 | 8.1 |
| AitZ (Zhang et al., 2024) | 1,987 | 6.0 |
| GUI Odyssey (Lu et al., 2025) | 7,735 | 15.3 |
| OS-Genesis (Sun et al., 2025) | 2,451 | 6.4 |
| WonderBread (Wornow et al., 2024) | 598 | 8.4 |
| AgentTrek (Xu et al., 2025a) | 10,398 | 12.1 |
| Mind2Web (Deng et al., 2023) | 2,350 | 7.3 |
| GUIAct (Chen et al., 2025) | 2,482 | 6.7 |
| AgentNet (Wang et al., 2025b) | 22,625 | 18.6 |
| **VideoAgentTrek** | **39,000+** | **39.25**[†] |

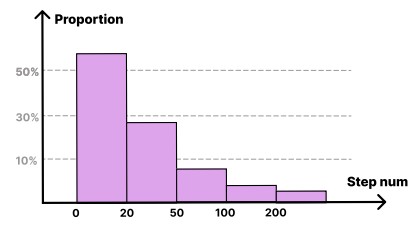

Table 13: VideoAgentTrek data distribution of step number

Table 12: Average steps across datasets (as reported in their papers). [†]Estimated from a 5,416-trajectory sample in our corpus.

and a manual inspection of 200 videos nearest to OSWorld clusters reveals no obvious task duplication, suggesting that VideoAgentTrek primarily provides diverse, generic computer-use supervision rather than narrow memorization of benchmark tasks, although residual overlap cannot be entirely ruled out.

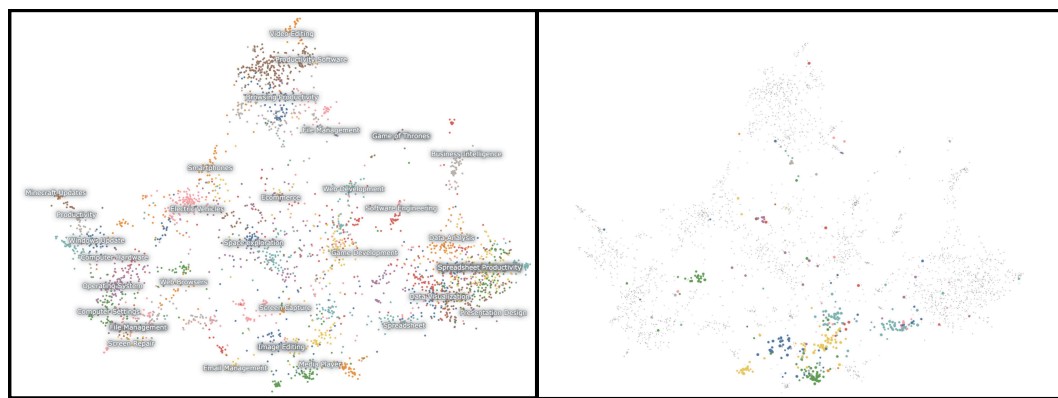

Figure 14: Text-embedding visualization of tutorial titles (gray) and OSWorld task instructions (colored) in a shared 2D space. Left: topic clusters of the tutorial corpus. Right: OSWorld instructions overlaid on the same distribution.

## J  LIMITAION AND FUTURE WORK

**Limitation.** VideoAgentTrek is currently constructed from 2D desktop screen recordings with predominantly English user interfaces. While this setting already covers a broad range of everyday computer-use workflows, it does not directly address mobile-only interaction patterns (for example, touch gestures and soft keyboards) or non-English UIs with different scripts, fonts, and themes. In addition, OSWorld-style benchmarks highlight that robust OCR and text-centric perception under diverse themes can further improve downstream performance. In this work, we intentionally focus on the video-to-action pipeline itself, that is, how to transform large-scale unlabeled screen-capture videos into high-quality pretraining trajectories, and rely on the underlying VLM/OCR backbone for text recognition, rather than explicitly optimizing OCR or resolution-specific components. In principle, the same pipeline can be extended beyond our current scope by adapting the screen filtering and action extraction modules (for example, replacing cursor-based filters with touch-indicator detection, enabling multilingual OCR/ASR, and incorporating mobile-specific GUI priors). Scaling

VideoAgentTrek to mobile platforms and non-English environments, together with more explicit OCR-enhanced variants, is an important direction for future work.

**Future Work**   Our work opens up several clear directions for future research. First, we would like to better understand the behavior of the current VideoAgentTrek pipeline itself, for example by running synthetic corruption studies (jittering click coordinates or corrupting a controlled fraction of parameters), performing more systematic data and model ablations (such as isolating the contribution of inner monologue), and exploring confidence-aware filtering to quantify and mitigate the impact of noisy supervision. Second, we aim to further improve the inverse dynamics module by going beyond purely visual signals, augmenting it with richer auxiliary cues such as ASR transcripts and keystroke audio, and potentially combining these with lightweight hybrid heuristics where they are reliable. Third, we plan to broaden the scope of VideoAgentTrek by extending the video-to-action pipeline to mobile platforms and non-English user interfaces, and by applying VideoAgentTrek pretraining to additional vision–language backbones when computational resources allow.

