# OpenReview forum: "VideoAgentTrek: Computer-Use Pretraining from Unlabeled Videos"
_ICLR.cc/2026/Conference — ICLR 2026 Poster_

### Official Review · Reviewer_c9gy · 2025-10-30

**Soundness:** 3
**Presentation:** 2
**Contribution:** 4
**Rating:** 8
**Confidence:** 3

**Summary:**

Paper introduces a scalable pipeline, VideoAgentTrek to automatically mine training data for use in training of computer-use agents.
The work aims to address the challenge costliness of manually annotating such training data at scale. It does so by exploiting the large amount of screen-recorded tutorials available online and introduces a means of extracting explicit structured action labels from such unlabeled videos.
To achieve this, the authors designed VIDEO2ACTION, a two component inverse dynamics module (IDM) consisting of a video grounding model and a action content recognizer.
The module first performs dense action-event detection to segment clips and assign action labels, then a action parameterization model analyzes these segments to produce structured parameters such as pointer coordinates, typed text etc.
Using 39000 YouTube tutorials, the pipeline generates 1.52M interaction steps for large-scale pretraining.

**Strengths:**

The scalability of the method, producing over a million structured steps providing a way to generate large amount of annotated data required for training.

Clear establishment of need for large scale data via performance scaling evaluation to support motivation and need for automated scalable data collection.

Inclusion of cognitive-style reasoning using inner monologue generation process to extract rationale for steps i.e explicit the intent, the local plan, and the expected state change, enhances model interpretability and could improve models reasoning.

**Weaknesses:**

It could be beneficial to show the performance across more vision-language models to see if the improvements are generalizable.

For the generation of inner monologue while effective, reliance on LLM-generated text raises concerns about consistency and reliability of the rationales. Were there any analysis on the outputs?
Quantitative human study on a small subset could provide some insight.

In the performance evaluation of Action Event Detection, what is the threshold for temporal overlap to count as a hit?

Minor comments not affecting rating:
Line 266: "ASR" define abbreviations\
line 376 typo "iff"

**Questions:**

Were the inner monologue used in the training?

In the first stage of action event detection, type a_k is predicted with the timestamps, why train the action parameterization model to again predict the type (line 245 - 247).\
it is mentioned in line 247 "when available, we optionally condition on the detector’s $a_k$ to stabilize type predictions.". How does this affect the performance of the parameterization model?

For the Stage 2 training, since it's stated that training was done on "curated set of clean, human-annotated trajectories" (line 310), were this the ones samples from Open CUA and AGUVIS.
If so how does the performance change if for stage 2, training data was drawn from only VideoAgentTrek generated trajectories and/or using a mix of human annotated and generated.

---

> ### Author Response · Authors · 2025-11-21
> **Official Comment by Authors (1/3)**
>
> Thank you very much for your thoughtful and positive review. We’re glad you appreciated both the scalability of our pipeline for generating large-scale annotated data, and the effectiveness of our experimental validation.
>
> We also noticed you have some constructive questions about our work, and we're happy to elaborate further below!
>
> ---
>
> > **W1:It could be beneficial to show the performance across more vision-language models to see if the improvements are generalizable.**
>
>
> **A:** We appreciate this suggestion and agree that training across multiple vision–language backbones would provide stronger evidence that our gains are broadly generalizable. However our training requires ~3,000 GPU hours per full run. Given resource and time constraints, we prioritized depth over breadth: comprehensive data ablations (0%/50%/100% data scaling, Stage-1 vs Stage-2 contributions, test-time scaling) and validation on two independent benchmarks. We appreciate this suggestion and are working on adding more backbones in our next version.
>
> At the same time, the primary goal of VideoAgentTrek is to introduce a scalable pipeline for extracting useful agent trajectories from unlabeled screen-capture videos. Our focus in this work is therefore to demonstrate that the data generated by this pipeline is genuinely beneficial for computer-use agents, and that the resulting improvements persist across different task suites and environments.
>
> Importantly, the **Video2Action** module is **model-agnostic**: it produces trajectories in standard (screenshot, thoughts, action with parameters) format consumable by any VLM. The consistent gains across both OSWorld-Verified (online, Ubuntu) and AgentNetBench (offline, Windows/macOS) suggests that the improvements are driven by the *data* rather than a particular model architecture or environment.
>
> We are also working on releasing ScreenFilter, Video2Action, the full data pipeline, and our complete video ID list with all manifests to support community validation across a wide range of backbones.
>
>
> ---
>
> > **W2:For the generation of inner monologue while effective, reliance on LLM-generated text raises concerns about consistency and reliability of the rationales. Were there any analysis on the outputs? Quantitative human study on a small subset could provide some insight.**
>
> **A:** We appreciate this concern and agree that the reliability of LLM-generated inner monologue is an important question.
>
> Our design follows the empirical evidence from AGUVIS and OpenCUA that step-wise, trajectory-conditioned CoT improves GUI agents’ grounding and end-to-end success. Building on those works, we use inner monologue as an explicit, supervised signal.
>
> Building on these findings, our inner-monologue pipeline in Video2Action is designed to **further improve consistency with actions and faithfulness** to the underlying tutorial.
>
> - We extend the OpenCUA-style framework by providing the CoT generator with the full **tutorial transcript around each action**, including segments before, during, and after the event, so that the generated inner monologue is grounded not only in multimodal observations and raw actions, but also in the narrator’s described intent, scene, and domain knowledge.
> - We additionally add a **review stage** that automatically checks whether the generated monologue is reasonable and aligned with the detected actions (for example, whether the described target and operation match the action type and the visible UI change); trajectories with clearly inconsistent or nonsensical monologue are filtered or regenerated.
>
> **Human study:** To further sanity-check reliability, we maintained a fixed trajectory validation subset spanning diverse scenarios (macOS Mail, Windows Calculator, macOS Preview, Chrome, etc.) and conducted a small-scale human inspection during development. Before launching large-scale CoT generation, we manually reviewed the outputs on this subset and found the inner monologue to be generally consistent with both the transcript and the action sequence; representative samples are now included in the appendix of the revised paper for transparency.
>
> **Human study:** To further sanity-check reliability, we conducted a small-scale human evaluation on 100 step-wise samples spanning diverse scenarios (macOS Mail, Windows Calculator, macOS Preview, Chrome, etc.). For each step, we checked whether the *action_description* matched our criteria. In total, **81%** of sampled steps were judged suitable for Stage-1 pretraining under these criteria, and we report the detailed breakdown in the appendix F.

---

> ### Author Response · Authors · 2025-11-21
> **Official Comment by Authors (2/3)**
>
> ---
>
> > **W3:In the performance evaluation of Action Event Detection, what is the threshold for temporal overlap to count as a hit?**
>
> **A:** Our action event detection evaluates **segment-level** predictions on a 0.5s time grid, with different overlap criteria for different action types:
>
> - For **short, instantaneous interactions** (e.g., `click`, `key`, `hotkey`, `scroll`, `doubleClick`, etc.), the action duration in the raw interaction log is very short and the exact delay between the input and the visible UI change is ambiguous. We therefore represent them with short segments aligned to the 0.5s grid and count a prediction as correct if the predicted segment **has any temporal overlap** with the ground-truth segment (i.e., non-zero intersection on the 0.5s grid).
>
> - For **longer interactions**, especially text-input actions such as `write/type`, we want the predicted segment to cover most of the actual typing episode, since the full span is important to capture the content being entered. In this case, we require the temporal overlap with the ground-truth segment to cover at least **80% of the ground-truth duration** (i.e., `|intersection| / |GT| ≥ 0.8`) for the detection to count as a hit.
>
> We have clarified these thresholds in the revised version and will release the evaluation code so that future work can reproduce and compare event grounding performance under the same temporal-overlap criteria.
>
>
>
> ---
>
> > **W4:Minor comments not affecting rating: Line 266: "ASR" define abbreviations, line 376 typo "iff"**
>
> **A:** We thank the reviewer for these careful suggestions. We have defined the abbreviation “ASR” at its first occurrence and corrected the typo “iff” on line 376 in the revised version.
>
>
> ---
>
> > **Q1:Were the inner monologue used in the training?**
>
> **A:** Yes, the inner monologue is explicitly used during training. Our motivation for introducing inner monologue is that tutorial **transcripts contain a large amount of useful information** that is not fully captured by raw pixels and low-level actions alone, including the narrator’s **intent** (“now let’s format this column”), high-level **domain knowledge** about the application, and **scene understanding** of what is currently shown on the screen. Encoding this information as step-wise CoT-style inner monologue allows the agent to learn these signals jointly with action execution, rather than discarding them.
>
>
> ---
>
> > **Q2:In the first stage of action event detection, type a_k is predicted with the timestamps, why train the action parameterization model to again predict the type (line 245 - 247).it is mentioned in line 247 "when available, we optionally condition on the detector’s to stabilize type predictions.". How does this affect the performance of the parameterization model?**
>
> **A:** Thank you for asking this valuable detail! Regarding why Stage 2 also predicts the **action type**: although Stage 1 already outputs a type \(a_k\) together with timestamps, we intentionally let the Stage-2 parameterization model jointly predict both the action type and its fine-grained parameters during training. Intuitively, asking the model to decide “what action is being taken” and “with which parameters” at the same time provides a richer supervision signal. In practice, we find that this joint prediction behaves more stably than a setup where Stage 2 only fills in parameters under a fixed type, especially for borderline cases such as drag vs.\ click or type vs.\ key combinations.
>
> For each detected event, we compare the type predicted by Stage 2 with the type proposed by Stage 1:
>
> - If the two types are **consistent**, we simply use the Stage-2 prediction, which already reflects the local visual and temporal context and yields a complete action specification. In practice, the two types agree in the vast majority of cases.
> - If they are **inconsistent**, we treat this as a potentially noisy case. We will condition on the action type proposed by stage 1 model and ask stage 2 model to predict full action with parameters. And then we use VLM (GPT-5) to compare the two candidate actions during post processing stage.
>
> We have added further implementation details of this procedure to the appendix for clarity.
>
> ---

---

> ### Author Response · Authors · 2025-11-21
> **Official Comment by Authors (3/3)**
>
> ---
>
> > **Q3:For the Stage 2 training, since it's stated that training was done on "curated set of clean, human-annotated trajectories" (line 310), were this the ones samples from Open CUA and AGUVIS. If so how does the performance change if for stage 2, training data was drawn from only VideoAgentTrek generated trajectories and/or using a mix of human annotated and generated.**
>
> **A:** Thank you for this important question about our training data design! We would like to clarify the distinct roles of different data sources:
>
> - Stage 1 Computer Use Pretraining with VideoAgentTrek: The automatically constructed trajectories from tutorial videos provide large-scale, diverse computer-use knowledge and long-horizon interaction sequences. However, these tutorials are not strictly goal-conditioned: they typically demonstrate multiple loosely related operations within a single video. For example, a video titled "How to use Excel functions" may cover various operations without a precise task instruction like "insert a new column in worksheet X and set it to currency format." This makes VideoAgentTrek data ideal for pretraining to introduce broad computer-use knowledge but insufficient for instruction-following alignment.
> - Stage 2 SFT with open-source data:  Our design goal for Stage 2 is to align the pre-trained model with goal-oriented, instruction-following behavior. Each trajectory in this stage is paired with a specific natural-language task instruction, and actions strictly follow that instruction to completion. We therefore use fully open-source OpenCUA and AGUVIS trajectories, which provide the goal-conditioned supervision necessary for instruction-following alignment.
>
> This two-stage design mirrors common LLM pretraining pipelines, where broad pretraining is followed by instruction-tuning. We have not yet systematically studied mixing non-goal-oriented VideoAgentTrek data with instruction-following SFT data, but we plan to explore such mixing strategies in future work to potentially improve both robustness and instruction-following capabilities.
>
>
>
> ----
>
>
> We thank Reviewer again for the encouraging review and insightful questions. Your comments helped us clarify how inner monologue is generated and used, specify evaluation details such as temporal overlap thresholds, and better articulate how Stage‑1 and Stage‑2 data play complementary roles. We hope these clarifications address your remaining concerns and are happy to elaborate further.

---

### Official Review · Reviewer_wmub · 2025-10-31

**Soundness:** 3
**Presentation:** 3
**Contribution:** 3
**Rating:** 8
**Confidence:** 3

**Summary:**

This paper presents VIDEOAGENTREK, a scalable pipeline that automatically extracts GUI-action trajectories from unlabeled screen-capture videos (39 k YouTube tutorials → 1.52 M steps) and uses them for large-scale pre-training of computer-use agents. A learned inverse-dynamics module (VIDEO2ACTION) first localizes actions in time (event detector, F1=0.78) and then infers their parameters (click coordinates, typed text, etc.; 65.8 % human-verified accuracy). Continued pre-training on the mined data followed by supervised fine-tuning improves task success on OSWorld-Verified from 9.3 % (SFT-only) to 15.8 % (+70 % relative) and step accuracy on AgentNetBench from 64.1 % to 69.3 %. The authors release SCREENFILTER and VIDEO2ACTION as open-source tools.

**Strengths:**

Novel, timely problem: Leveraging the enormous volume of passive screen-capture videos for GUI-agent training is an appealing idea that addresses the current data bottleneck.
End-to-end pipeline: From raw YouTube crawl to executable (screenshot, action, parameters) tuples, the system is fully automated and scales to web size.
Strong empirical gains: Clear, statistically meaningful improvements over a pure SFT baseline on two independent benchmarks, plus positive scaling curves with data volume and test-time compute.

**Weaknesses:**

See questions.

**Questions:**

This paper presents a timely and impactful contribution by introducing VIDEOAGENTREK, the first fully-automated pipeline that converts unlabeled, publicly available screen-capture videos into large-scale, training-ready trajectories for GUI agents. By equipping an inverse-dynamics module (VIDEO2ACTION) with dense event detection and parameter extraction, the authors bypass the expensive manual-annotation bottleneck and demonstrate clear downstream gains on both online and offline benchmarks.
One open question remains: will the complete codebase (SCREENFILTER, VIDEO2ACTION training & inference scripts, data-preparation pipeline) and the processed VideoAgentTrek dataset be publicly released?

---

> ### Author Response · Authors · 2025-11-21
>
> Thank you very much for the positive and careful review. We’re happy that you see our work as a timely way to ease the data bottleneck for computer-use agents. We also appreciate that you highlighted both the end-to-end nature of our pipeline and the clear gains over the SFT-only baseline, as well as the good scaling behavior.
>
> We also noticed your open question about release plans, and we’re happy to share the details in the response below.
>
> ---
>
> > **Q:This paper presents a timely and impactful contribution by introducing VIDEOAGENTREK, the first fully-automated pipeline that converts unlabeled, publicly available screen-capture videos into large-scale, training-ready trajectories for GUI agents. By equipping an inverse-dynamics module (VIDEO2ACTION) with dense event detection and parameter extraction, the authors bypass the expensive manual-annotation bottleneck and demonstrate clear downstream gains on both online and offline benchmarks. One open question remains: will the complete codebase (SCREENFILTER, VIDEO2ACTION training & inference scripts, data-preparation pipeline) and the processed VideoAgentTrek dataset be publicly released?**
>
>
> **A:** Thank you for this important feedback. We fully understand that data availability is crucial for reproducibility and community impact. However, because our corpus is mined from YouTube, we must comply with the platform’s terms of service and therefore cannot redistribute the raw videos or audio directly. Within these legal and ethical constraints, we are strongly committed to maximizing what we can share.
>
>
> What we will release:
>
> 1. Complete toolkits: Full source code and pretrained weights for both ScreenFilter and Video2Action, enabling the community to apply our pipeline to their own video sources
> 2. Dataset manifest: A comprehensive list including:
>
> - YouTube Video IDs for all 39,000 processed videos
> - Full action metadata for all 1.52M steps (action types, parameters, thoughts, timestamps)
>
> 3. Reconstruction script: An automated downloader and preprocessor that reconstructs the exact training corpus from the manifest
>
> 4. CC‑licensed subset: We verify videos under a Creative Commons license, allowing us to release their trajectories along with full screenshots and metadata. This produces an immediately usable dataset for validation.
>
>
> We believe this release plan balances reproducibility with legal compliance, and provides the community with everything needed to replicate our work and extend it to new video sources. We are working on the release and  will make this release explicit in the camera-ready version.

---

### Official Review · Reviewer_b4VU · 2025-11-01

**Soundness:** 3
**Presentation:** 3
**Contribution:** 2
**Rating:** 4
**Confidence:** 4

**Summary:**

The paper introduces VideoAgentTrek, which synthesizes GUI interaction trajectories from actionless videos to replace costly manual annotations. The core is the Video2Action inverse dynamics module: it first localizes events, then infers action parameters, and finally synthesizes a chain-of-thought. Training proceeds in two stages: supervised fine-tuning on the full dataset, followed by fine-tuning on a human-verified subset. Experiments report absolute gains of +6% on OSWorld-Verified and +5% on AgentNetBench. The authors release the ScreenFilter and Video2Action tools but do not release the video dataset or the trajectory dataset.

**Strengths:**

- The paper is easy to follow. The data processing pipeline, training setup, and results are clearly presented.
- Applying VPT-style ideas to GUI agents is intuitive and scalable for data collection, and the idea is well-executed.
- Clear improvements on two popular benchmarks.
- ScreenFilter and Video2Action are released to support the reproduction of the video annotation pipeline.

**Weaknesses:**

- While tools are open-sourced, the full video corpus and trajectory annotations are not available, which I believe could significantly increase the contribution of this work.
- Potential data leakage is not quantified. The training data derived from public tutorials may overlap with OSWorld/AgentNetBench tasks, but the paper does not appear to include a rigorous deduplication or leakage analysis.

**Questions:**

- Tutorial videos often contain a lot of noise (e.g., extraneous/meaningless mouse movements, hotkey usage). Is there any specific mechanism to handle this? For text-input actions, how does the model differentiate between a string-level typing action and individual key presses?
- How are frames with no user action but visible UI changes handled? Does the Video2Action model have an explicit noop action?
- Cursor icons can vary across operating systems and applications. What cursor types are supported by the cursor detection model?
- Do event detection and cursor detection models require a standardized resolution/frame rate? How robust is the system across varying FPS and resolutions?
- Can you provide an ablation on the effect of automatically generated inner monologue?
- Is there a plan to release the full video dataset and extracted trajectories? Public release, especially of high-resolution videos with audio, would substantially strengthen the paper’s contribution. I would increase my score if this were to become available.

---

> ### Author Response · Authors · 2025-11-21
> **Official Comment by Authors (1/3)**
>
> Thank you for your careful review and for recognizing the intuitive appeal and strong execution of our VPT-style approach to GUI agents. We appreciate your acknowledgment of our clear improvements on both benchmarks and the value of releasing ScreenFilter and Video2Action tools.
>
> ---
>
> > **W1:While tools are open-sourced, the full video corpus and trajectory annotations are not available, which I believe could significantly increase the contribution of this work.**
>
> **A:** We fully agree that releasing data and tools is crucial for reproducibility and impact, and we appreciate you highlighting this. Because our corpus is mined from YouTube, we **cannot redistribute raw videos or audio** due to platform terms of service. However, we are committed to releasing everything we *can* within these constraints, and we are actively working on the following:
>
> 1. Complete toolkits: Full source code and pretrained weights for both ScreenFilter and Video2Action, enabling the community to apply our pipeline to their own video sources
> 2. Dataset manifest: A comprehensive list including:
>
> - YouTube Video IDs for all 39,000 processed videos
> - Full action metadata for all 1.52M steps (action types, parameters, thoughts, timestamps)
>
> 3. Reconstruction script: An automated downloader and preprocessor that reconstructs the exact training corpus from the manifest
>
> 4. CC‑licensed subset: We verify videos under a Creative Commons license, allowing us to release their trajectories along with full screenshots and metadata. This produces an immediately usable dataset for validation.
>
>
> We believe this release plan balances reproducibility with legal compliance, and provides the community with everything needed to replicate our work and extend it to new video sources. We are working on the release and  will make this commitment explicit in the camera-ready version.
>
> ---
>
> > **W2:Potential data leakage is not quantified. The training data derived from public tutorials may overlap with OSWorld/AgentNetBench tasks, but the paper does not appear to include a rigorous deduplication or leakage analysis.**
>
> **A:** We appreciate this concern and agree that potential contamination between pretraining data and evaluation benchmarks must be taken seriously. Conceptually, there is a significant mismatch between our **long-form tutorial videos** and the **goal-conditioned tasks** in OSWorld and AgentNetBench:
>
> - Tutorials focus on *demonstrating features* of applications on ad-hoc content (e.g., showing several Excel functions on a synthetic spreadsheet), usually without a single, precise end goal.
>
> - OSWorld and AgentNetBench define *concise, instruction-driven* tasks with clearly specified target states (e.g., opening a particular file, configuring a specific setting).
>
> Thus, even when the same application appears in both sources, the tutorial data mostly provides **generic interaction patterns** rather than direct solutions to benchmark tasks.
>
> To make this gap more concrete, we have added a **“Benchmark contamination analysis”** section in the appendix(Appendix I). In this analysis:
>
> - We encode **all tutorial video titles** and **all OSWorld task instructions** into a shared text-embedding space.
> - We project these embeddings into 2D (Figure 14) and observe that:
>   - OSWorld instructions form several relatively compact clusters, and
>   - Our tutorial titles are widely dispersed across the embedding space, covering many disparate domains and workflows.
> - We also confirm that we **do not intentionally scrape OSWorld/AgentNetBench solution videos** and find no explicit mentions of “OSWorld” or “AgentNetBench” in titles or transcripts.
> - We manually inspect **200 tutorial videos** whose embedded titles are closest to the OSWorld clusters. These nearest neighbors are generic “how to use [application]” tutorials rather than benchmark-specific solutions; we do **not** find obvious duplication of OSWorld/AgentNetBench tasks.
>
> This evidence supports our claim that VideoAgentTrek primarily supplies generic interaction experiences and domain knowledge, not direct memorization of benchmark tasks.

---

> ### Author Response · Authors · 2025-11-21
> **Official Comment by Authors (2/3)**
>
> ---
>
> > **Q1:Tutorial videos often contain a lot of noise (e.g., extraneous/meaningless mouse movements, hotkey usage). Is there any specific mechanism to handle this? For text-input actions, how does the model differentiate between a string-level typing action and individual key presses?**
>
> **A:** We agree that tutorial videos contain substantial interaction “noise” such as small cursor jitters or incidental key presses. However, we do **not** treat all of these as harmful: as long as an interaction leads to a meaningful GUI state transition (e.g., opening a menu, confirming a dialog, editing a field), it provides useful “experience” for the model to learn realistic state transitions and application usage patterns, especially when combined with inner monologue derived from transcripts.
>
> Our design is naturally tolerant to such noise because of how events are defined and learned. The inverse-dynamics model is trained on **real human interaction logs** where events are already defined at a **semantic action level** (clicks, drags, typing segments, etc.), not at the level of every micro-movement of the cursor. Many small, aimless cursor wiggles or inconsequential key taps never appear as labeled events because they do not correspond to deliberate, state-changing actions in the logs. As a result, when we apply the trained VIDEO2ACTION model to tutorial videos, it inherits this bias toward **salient, state-changing interactions**: in practice, it predicts actions like opening menus, confirming dialogs, and editing fields, while low-level motion noise is absorbed into the visual context between events rather than materialized as separate steps in the trajectory.
>
> For **text input**, we follow the keyboard schema defined in the AgentNet data. In the real user logs, **consecutive printable character key presses are aggregated into one string-level typing action**, represented as `type("...")`. Video2Action is trained directly in this **string-level action space**, so at inference time it predicts **string-typed actions for content input**, rather than a long sequence of per-character key presses.
>
> Thus, text-input actions are naturally represented as single `type` actions, which is the granularity used throughout our training and evaluation.
>
>
> ---
>
> > **Q2:How are frames with no user action but visible UI changes handled? Does the Video2Action model have an explicit noop action?**
>
> **A:** This is an important practical scenario (e.g., loading animations, background notifications, auto-refreshing content). Our design handles such cases via how **events are defined and predicted**, rather than through an explicit `noop` token.
>
> - In the **training data**, events are aligned with ground-truth mouse/keyboard inputs recorded by annotation tool. Frames in which the UI changes *without* any corresponding user input (like automatic pop-ups or progress bars) simply **receive no event label**.
> - Video2Action is trained to map frame sequences to a **set of event timestamps and actions**. Intervals without labeled user input are treated as context and do *not* produce a separate action step.
>
> At **inference time**, the model similarly outputs **discrete event timestamps** along with their action parameters. Frames or intervals for which no event is predicted are implicitly treated as **no-op**; we do not introduce an explicit `noop` action type in the action space. UI-only changes without user input thus appear only as visual context between two detected events, rather than as trajectory steps.
>
> ---
>
> > **Q3:Cursor icons can vary across operating systems and applications. What cursor types are supported by the cursor detection model?**
>
> **A:** Our cursor detection model is trained to recognize **58 commonly used cursor types** across operating systems and applications. These include, for example:
>
> - Default pointers (arrows) in Windows and macOS,
> - Text-selection I-beams,
> - Hand cursors for clickable elements,
> - Resize and move cursors,
> - Loading/busy cursors and other standard system shapes.
>
> We provide a detailed enumeration of these cursor types and their frequencies in **Table 3**. This set is designed to cover the most frequent cursor variants seen in our tutorial corpus on mainstream desktop OSes.

---

> ### Author Response · Authors · 2025-11-21
> **Official Comment by Authors (3/3)**
>
> ---
>
> > **Q4:Do event detection and cursor detection models require a standardized resolution/frame rate? How robust is the system across varying FPS and resolutions?**
>
> **A:** Our pipeline is designed to handle **different resolutions and original frame rates** by normalizing inputs at the pre-processing and model levels.
>
> - **Frame rate.**
>   All videos are normalized to a standard frame rate (30 FPS) before processing. We then use **fixed temporal sampling strategies**:
>
>   - Cursor detection: sampling at **1 FPS**
>   - Event detection: sampling at **4 FPS**
>
>   This reduces the impact of variable original FPS and ensures that both modules operate on a consistent temporal grid.
>
> - **Resolution.**
>   Our dense event detection training data spans a range of resolutions, summarized in **Table 5** (e.g., large clusters around 1280×720 and related aspect ratios, plus higher resolutions such as 2560×1440 and a tail of “other” resolutions).
>
>   - For **cursor detection**, we use YOLOv8x, which resizes input frames to **640×640** while preserving aspect ratio. This gives the model a normalized input size and naturally accommodates different original resolutions.
>   - For **event detection**, we use Qwen2.5-VL, which adopts a NaViT-style architecture that operates directly on images at their native resolution, so it does not require a fixed resolution during inference.
>
> In practice, because the training set already covers a variety of resolutions (see Table 5) and all videos are normalized in frame rate before sampling, we designed the system to be robust across typical desktop recording conditions. We have clarified these implementation details and the resolution distribution in the paper, and a more detailed, stratified breakdown of performance versus resolution/FPS would be a valuable direction for extended analysis.
>
> ---
>
> > **Q5:Can you provide an ablation on the effect of automatically generated inner monologue?**
>
> **A:** We agree that a direct ablation on inner monologue would be valuable. At present, we have not run a full-scale ablation on our own corpus. The main practical constraint is that Stage-1 pretraining at our scale requires roughly 3,000 GPU hours per run, so rerunning the entire pipeline with and without inner monologue is beyond our current compute budget within the review timeline.
>
> However, our choice to include automatically generated inner monologue is not arbitrary. It is motivated both by the information of our data and by prior empirical evidence:
>
> - Our tutorial videos come with **rich narration transcripts**, where instructors frequently articulate their intent (“now I’ll format this as currency”), describe the current state, and explain application concepts. Encoding this information as step-wise chain-of-thought / inner monologue allows the agent to jointly learn goal understanding, screen interpretation, and action execution rather than relying solely on pixel transitions.
> - Recent work has already studied the impact of inner monologue on GUI/computer-use agents:
>   - **AGUVIS** (Xu et al., 2024) reports that removing inner monologue leads to consistent performance drops across both GUI grounding benchmarks and low-level control tasks.
>   - **OpenCUA** (Wang et al., 2025) conducts ablations over different chain-of-thought formats and finds that richer, reflective reasoning traces significantly improve success rates on OSWorld and related benchmarks.
>
> These results support our design decision to incorporate inner monologue when mining trajectories from narrated tutorials.
>
> ---
>
> > **Q6:Is there a plan to release the full video dataset and extracted trajectories? Public release, especially of high-resolution videos with audio, would substantially strengthen the paper’s contribution. I would increase my score if this were to become available.**
>
> **A:** As discussed in **W1**, we are strongly committed to maximizing what we can release while complying with YouTube’s terms of service. We are actively working on preparing the manifest, reconstruction scripts, and CC subset and intend to release them as soon as they are ready. We will clarify this release plan explicitly in the paper. We hope this addresses your concern about data availability and strengthens the contribution from a reproducibility and community impact perspective.
>
> ------
>
> Once again, we thank you for your thoughtful review and for highlighting both the strengths and the key areas where additional clarification and analysis are important.

---

### Official Review · Reviewer_Rbqb · 2025-11-01

**Soundness:** 3
**Presentation:** 3
**Contribution:** 3
**Rating:** 6
**Confidence:** 4

**Summary:**

The paper proposes VideoAgentTrek, a scalable pipeline to pretrain computer-use agents from unlabeled, publicly available screen-recorded videos. The key challenge is that raw videos  lack structured action labels (types, timestamps, and parameters). The authors address this by introducing an inverse dynamics module to generate the parameterized action tuples. With the collected dataset and the proposed multi-stage training strategy, the final performance of the GUI Agent achieved SOTA.

**Strengths:**

1. Interesting and reasonable reframing of GUI action recovery as inverse dynamics from raw videos without (great) manual labels.

2. The filter system leverages cursor detection to automatically focus on GUI-heavy segments; channel-coherence expansion for source discovery is practical and scalable.

3. Clear improvements over a strong SFT-only baseline on both OSWorld-Verified and AgentNetBench, plus analysis showing benefits grow with pretraining scale and planning horizon.

4. Pipeline is well-structured and explained (collection → filtering → detection → parameterization → rationale → training).

5. Practical tools and release plans (if they actually release finally) can catalyze broader community work.

**Weaknesses:**

1. While Stage 2 SFT mitigates noise (Parameterization accuracy, detection errors, etc), a more systematic study quantifying how parameter errors propagate to agent performance (e.g., via controlled corruption of parameters) would clarify robustness and failure modes.

2. The dense detector shows weaker recall for keyboard/press actions, which can be crucial in many workflows. It would help to analyze how missing non-pointer actions affects task categories (e.g., auth flows, terminal usage) and to explore augmentations (ASR cues, keystroke audio hints) to close this gap. Also, since the author uses the pointer as one of the  video filters, I am very curious whether the trained model would bias to such pattern? Further discussion and analysis would help.

3. More detailed release of manifests (video IDs, timestamps, filtering decisions) would further improve reproducibility within ToS boundaries.

4.  The pipeline currently focuses on 2D screen recordings. Some OSWorld tasks involve subtle text/element detection under variable themes or require OCR robustness. An ablation with explicit OCR enhancements or higher-res sampling could clarify performance ceilings. Similarly, mobile platforms or non-English UIs are out-of-scope; discussing adaptation strategies would strengthen the broader impact story.

5. In Table 9, the authors provided a cross-dataset comparison, which is good; But I do think there misses some important datasets. I would prefer a more complete comparison as done in AgentNet, Table 2. This will help the readers to localize.

6. From the Figure, it seems the "test-time scaling" helps a lot, but I cannot find the further details.

**Questions:**

1. See the weakness.

2. Have you tried ensembling multiple parameter predictors or using confidence-based filtering to drop uncertain steps? How does selectively pruning low-confidence parameterizations affect Stage-1 benefits?

3. Can you quantify how coordinate noise (e.g., ±k pixels) affects downstream success on benchmarks via a controlled perturbation study?
Improving non-pointer actions:

4. For press/type, did you explore fusing ASR transcripts or keystroke sounds to improve recall? Are there simple heuristics (e.g., stable cursor + text change) that boost detection without heavy supervision?


5. Beyond releasing filter tools, can you also share:
- Video ID lists and segment timestamps (without frames), with pass/fail reason codes.
- A small, license-cleared demo subset (e.g., Creative Commons) to enable exact replication of Stage-1 on a miniature scale.

6. Did video-pretraining improve performance uniformly across OSWorld app buckets (calc, chrome, vscode, etc.), or are gains concentrated in certain domains?

7. Since mined text can include sensitive inputs (typed passwords, emails), how to deal with?

8. Could you incorporate UI element detection (e.g., layout parsing) to normalize parameterization into element-centric actions rather than screen coordinates, improving cross-resolution robustness?

---

> ### Author Response · Authors · 2025-11-21
> **Official Comment by Authors (1/5)**
>
> Thank you for your thorough and constructive review. We greatly appreciate that you recognized the soundness of our inverse dynamics approach, the scalability of our pipeline, and the clear improvements over strong baselines.
>
> ---
>
> > **W1:While Stage 2 SFT mitigates noise (Parameterization accuracy, detection errors, etc), a more systematic study quantifying how parameter errors propagate to agent performance (e.g., via controlled corruption of parameters) would clarify robustness and failure modes.**
>
> **A:** We agree that understanding how noisy parameter supervision propagates to downstream behavior is important. Our Stage‑1 video continued pretraining is designed to enrich the model’s long‑horizon planning and computer‑use knowledge, but it is based on automatically extracted actions that are not perfectly clean. This noise is most likely to affect fine‑grained execution skills such as GUI grounding.
>
> To quantitatively probe this, we further evaluate all model variants on **OSWorld‑G**, which directly measures the low-level grounding ability to localize and select the correct on‑screen target. The results combined with the OSWorld and AgentNetBench (also reported in the paper(Table 6)) are:
>
> | Model variant             | OSWorld-G | OSWorld | AgentNet Bench |
> | ------------------------- | --------- | ------- | -------------- |
> | Qwen2.5-VL-7B             | 31.40     | 4.5     | 38.5           |
> | + Stage-2 SFT only        | 31.56     | 9.3     | 64.1           |
> | + Stage-1 CPT only        | 26.24     | -       | -              |
> | Stage-1 CPT + Stage-2 SFT | 30.50     | 15.78   | 69.3           |
>
> These results indicate that:
>
> 1. **Effect of noisy Stage‑1 supervision.** Stage‑1 CPT alone reduces OSWorld‑G from 31.40 to 26.24, indicating that training directly on noisy trajectories can indeed slightly shift low‑level grounding.
> 2. **Mitigation via Stage‑2 SFT.** After Stage‑2 SFT on clean, curated GUI‑agent demonstrations, OSWorld‑G almost returns to the base level (30.50 vs. 31.40), while OSWorld and AgentNetBench improve substantially compared to both the base model and the Stage‑2‑only model.
>
> This shows that noisy Stage‑1 supervision does have a measurable effect on a pure grounding benchmark, but our two‑stage design keeps this effect limited while enabling large gains in end‑to‑end computer‑use performance.
>
> A dedicated synthetic corruption study (e.g., progressively jittering click coordinates by ±k pixels or corrupting a controlled fraction of parameters) is indeed a very valuable direction to further characterize robustness and failure modes. Due to time and resource constraints during the review period, we did not run such a dedicated perturbation study, but this naturally extends our OSWorld-G analysis, and we plan to explore it further in future work (Appendix I).
>
> ---

---

> ### Author Response · Authors · 2025-11-21
> **Official Comment by Authors (2/5)**
>
> ---
>
> > **W2:The dense detector shows weaker recall for keyboard/press actions, which can be crucial in many workflows. It would help to analyze how missing non-pointer actions affects task categories (e.g., auth flows, terminal usage) and to explore augmentations (ASR cues, keystroke audio hints) to close this gap. Also, since the author uses the pointer as one of the video filters, I am very curious whether the trained model would bias to such pattern? Further discussion and analysis would help.**
>
> **A:** **Non-pointer actions and keyboard recall.** We fully agree that visual detection of keyboard events is intrinsically challenging and that non‑pointer actions are important for workflows involving shortcuts, terminal usage, or rich text editing. In the paper, we already report action‑type‑wise dense detection metrics (Table 1) : **type** events (text content input) are detected reasonably well, whereas **press** events (shortcut‑like key presses) indeed have low recall. Importantly, press actions are relatively rare in our corpus: they constitute only 4.15% labeled actions. In practice, we find that many real‑world workflows can be completed with point‑and‑click plus typing, so missing part of the press‑only signal is less catastrophic than one might expect. Nonetheless, this is a clear limitation of the current visual‑only detector.
>
> On the downstream side, the OSWorld‑Verified per‑application results (Table 7, partially reproduced below) suggest that video pretraining benefits are not limited to pointer‑dominated tasks. Under the 50‑step budget, comparing Stage‑2‑only vs. Stage‑1+Stage‑2:
>
> - **VSCode** (keyboard‑heavy): solved tasks increase from 13.0% to 26.1%.
> - **Writer:** from 8.7% to 26.1%.
> - **Chrome:** from 26.1% to 32.6%.
> - **Workflow bucket:** from 4.3% to 8.7%.
>
> This indicates that despite imperfect press detection, Stage‑1 video pretraining still improves performance in domains where keyboard actions are important, and does not systematically harm them.
>
> We agree that integrating **ASR cues** or **keystroke audio** (where available) could be very helpful for closing the gap on press‑type actions, as the reviewer suggests. Our current action detection system does not yet fuse such signals into the detector, but we view this as a promising extension and will discuss it explicitly as future work.
>
> **Pointer‑based filtering and potential bias.** Our ScreenFilter detects a broad range of cursor states (arrow, hand, caret/I‑beam, loading spinner, etc.) that are ubiquitous in real desktop GUIs(Table 3). In typical desktop workflows, the cursor remains visible even during primarily keyboard‑driven interactions. As a result, filtering for visible pointer states generally aligns the training distribution with realistic GUI usage while excluding unrelated content, such as full presenter‑face shots or tutorial transitions.  During development, this proved especially useful: a lightweight YOLO model could reliably filter out irrelevant content early on while maintaining high precision and recall (91% and 89%) on the held-out test set. Moreover we retains video segments where at least 80% frames contains cursor to tolerate potiential missing and improve recall. We also reflect these details in Appendix C.
>
> ---
>
> > **W3:More detailed release of manifests (video IDs, timestamps, filtering decisions) would further improve reproducibility within ToS boundaries.**
>
> **A:** Thank you for this important feedback. We fully understand that data availability is crucial for reproducibility and community impact, and we are strongly committed to maximizing what we can share within legal and ethical constraints.
>
> What we will release:
>
> 1. Complete toolkits: Full source code and pretrained weights for both ScreenFilter and Video2Action, enabling the community to apply our pipeline to their own video sources
> 2. Dataset manifest: A comprehensive list including:
>
> - YouTube Video IDs for all 39,000 processed videos
> - Full action metadata for all 1.52M steps (action types, parameters, thoughts, timestamps)
>
> 3. Reconstruction script: An automated downloader and preprocessor that reconstructs the exact training corpus from the manifest
>
> 4. CC‑licensed subset: We verify videos under a Creative Commons license, allowing us to release their trajectories along with full screenshots and metadata. This produces an immediately usable dataset for validation.
>
>
> We believe this release plan balances reproducibility with legal compliance, and provides the community with everything needed to replicate our work and extend it to new video sources. We are working on the release and  will make this commitment explicit in the camera-ready version.

---

> ### Author Response · Authors · 2025-11-21
> **Official Comment by Authors (3/5)**
>
> ---
>
> > **W4:The pipeline currently focuses on 2D screen recordings. Some OSWorld tasks involve subtle text/element detection under variable themes or require OCR robustness. An ablation with explicit OCR enhancements or higher-res sampling could clarify performance ceilings. Similarly, mobile platforms or non-English UIs are out-of-scope; discussing adaptation strategies would strengthen the broader impact story.**
>
> **A:** We agree that GUI interaction is heavily text‑centric and that robust OCR (across themes, fonts, and languages) can further improve downstream performance. Our work is primarily focused on the **video‑to‑action pipeline** : how to convert large‑scale, unlabeled computer‑use videos into useful trajectories for agent pretraining, rather than on improving the underlying OCR or visual backbone per se. The backbone can, in principle, benefit from any OCR pretraining data or higher‑resolution visual modeling, and our pipeline is compatible with such improvements.
>
> In the current submission we restrict ourselves to desktop, which we now emphasize more clearly as a limitation. Extending VideoAgentTrek to mobile settings is conceptually straightforward but practically requires adapting:
>
> - the screen filtering to mobile‑specific signals (touch indicators, platform‑specific UI chrome),
> - the action extraction to mobile interaction patterns (taps, long‑presses, gestures),
>
> We have expanded the limitations and broader-impact discussion to explicitly cover these points and outlined how the same video-to-action approach can be adapted to mobile environments(Appendix J).
>
>
> ---
>
> > **W5:In Table 9, the authors provided a cross-dataset comparison, which is good; But I do think there misses some important datasets. I would prefer a more complete comparison as done in AgentNet, Table 2. This will help the readers to localize.**
>
> **A:** Thank you for this suggestion. We agree that a more exhaustive comparison is very helpful for readers to localize our contribution. In the revised version, we expand the cross‑dataset table (Table 12 in the appendix) to include additional relevant computer‑use datasets following the style of AgentNet’s Table 2 (e.g., web automation and GUI datasets). The updated table clarifies where VideoAgentTrek fits in terms of scale and domain coverage.
>
>
> ---
>
> > **W6: From the Figure, it seems the "test-time scaling" helps a lot, but I cannot find the further details.**
>
> **A:** In our work, “test‑time scaling” refers specifically to increasing the **allowed action‑step budget during evaluation**.
>
> On OSWorld‑Verified, we evaluate the same agent under two budgets: 20 vs. 50 allowed actions. As shown in Figure 5 (and discussed in Sec. 4.1 / Sec. 4.2),  Under the 20‑step setting, the Stage‑2‑only baseline shows no benefit from a larger budget. In contrast, the Stage‑1+Stage‑2 agent continues to improve when moved from 20 to 50 steps, yielding an additional relative gain of ~11.7% on top of its already stronger base performance. This behavior is consistent across OSWorld‑Verified app buckets (also reflected in Table 7).
>
> | OSWorld‑Verified      | SR@20 | SR@50 |
> | --------------------- | ----- | ----- |
> | Stage-2-only agent    | 9.3   | 9.2   |
> | Stage‑1+Stage‑2 agent | 14.13 | 15.78 |
>
> We attribute this to the **long‑horizon supervision** provided by VideoAgentTrek: our mined trajectories have an average length of 39.25 steps, with 42.1% exceeding 20 steps and 14.5% containing at least 50 steps. Exposure to such long trajectories during Stage‑1 pretraining appears to teach the model to make productive use of larger step budgets at test time, instead of simply taking more random or redundant steps. We will add a clearer explanation of this setup to the main text(section 3.1.2).
>
> ---
>
> > **Q2:Have you tried ensembling multiple parameter predictors or using confidence-based filtering to drop uncertain steps? How does selectively pruning low-confidence parameterizations affect Stage-1 benefits?**
>
> **A:** We have not yet explored ensembling or confidence-based pruning of parameter predictions in this work. We agree that confidence-aware filtering of low-quality steps is a promising way to improve the scale–vs–noise trade-off in Stage-1, and we will highlight this as an avenue for future work(Appendix J).
>
> As discussed in W1, noisy Stage-1 supervision has a measurable but limited impact on pure grounding metrics, and our two-stage scheme (Stage-1 CPT followed by Stage-2 SFT on clean demonstrations) largely recovers grounding performance while yielding large gains in end-to-end computer-use benchmarks. We expect that selectively pruning low-confidence parameterizations could further reduce the residual impact of Stage-1 noise on grounding and execution, while preserving—and potentially amplifying, the long-horizon benefits brought by large-scale video pretraining.

---

> ### Author Response · Authors · 2025-11-21
> **Official Comment by Authors (4/5)**
>
> ---
>
> > **Q3:Can you quantify how coordinate noise (e.g., ±k pixels) affects downstream success on benchmarks via a controlled perturbation study? Improving non-pointer actions:**
>
> **A:** This is closely related to W1, and we fully agree that a controlled perturbation study, explicitly injecting coordinate noise (e.g., ±k pixels) and measuring its impact on success rates—would provide a very clear picture of robustness to spatial errors.
>
> In the current submission, we did not run such a dedicated perturbation experiment, but analyze robustness by adding OSWorld‑G evaluation, which isolates grounding performance and reveals how noisy Stage‑1 supervision affects low‑level pointing accuracy (please see W1).
>
> This analysis indicates that parameter noise does impact grounding, but the two‑stage training scheme keeps this impact contained while enabling large gains in end‑to‑end performance. A focused coordinate‑noise ablation is an excellent suggestion, due to resource and time constraints during the review period we were not able to fullly study it, but it's a natural extensions of our OSWorld‑G evaluation study and we will further explore this in the revised version.
>
> ---
>
> > **Q4:For press/type, did you explore fusing ASR transcripts or keystroke sounds to improve recall? Are there simple heuristics (e.g., stable cursor + text change) that boost detection without heavy supervision?**
>
> **A:** In our current system, ASR transcripts are used to generate **inner monologue** for Stage‑1 (as described in the paper), and are not directly fused into the dense action detector. Keystroke audio is often suppressed in creator setups (noise cancellation, external microphones), so it is not a reliable signal in our collected videos.
>
> ---
>
> We agree that richer multimodal cues—e.g., ASR transcript alignment for type/press events, or explicit keystroke sounds when available, could substantially help improve recall of non‑pointer actions, especially shortcut‑like **press** events. We will highlight this as an important extension of Video2Action.
>
> Regarding heuristic rules such as “stable cursor + localized text change,” we have found such rules to be brittle at scale: in web videos, many visual changes are not directly linked to user input (automatic refreshes, pop‑ups, notifications), which can lead to many false positives when applied blindly across tens of thousands of hours of footage. Our design choice was therefore to rely on a learned dense detector and to focus on improving it via more data and better supervision, rather than enumerating heuristics. Nonetheless, hybrid heuristics + learned models are an interesting direction, and we have mentioned this explicitly as potential future work(Appendix J).
>
> ---
>
> > **Q5:Beyond releasing filter tools, can you also share:Video ID lists and segment timestamps (without frames), with pass/fail reason codes.A small, license-cleared demo subset (e.g., Creative Commons) to enable exact replication of Stage-1 on a miniature scale.**
>
> **A:** This is closely related to W3, and we fully agree. We will make these plans explicit in the paper and on the project repository so that others can reproduce and extend our work.
>
>
> ---
>
> > **Q6:Did video-pretraining improve performance uniformly across OSWorld app buckets (calc, chrome, vscode, etc.), or are gains concentrated in certain domains?**
>
> **Response:**
> The OSWorld‑Verified per‑application results in Appendix Table 7 are intended to address this question. Comparing Stage‑2‑only vs. Stage‑1+Stage‑2 under the same 50‑step budget:
>
> | Model           | Task SR (%) | Calc | Chrome | GIMP | VSCode | Writer | ThunderBird | OS    | Impress | Workflow | VLC   |
> | --------------- | ----------- | ---- | ------ | ---- | ------ | ------ | ----------- | ----- | ------- | -------- | ----- |
> | stage2 only     | 9.27        | 2.2% | 26.1%  | 7.7% | 13.0%  | 8.7%   | 13.3%       | 8.3%  | 8.5%    | 4.3%     | 5.9%  |
> | stage1 + stage2 | **15.78**   | 2.2% | 32.6%  | 7.7% | 26.1%  | 26.1%  | 40%         | 16.7% | 12.8%   | 8.7%     | 17.6% |
>
> We see that video pretraining Improves most domains while at worst keeping performance unchanged (e.g., GIMP and Calc). Thus, the gains are **broadly distributed across app buckets** and not concentrated in a single domain. We will make this cross‑domain improvement more explicit in the main text(Section 3.1.2).
>
> ---

---

> ### Author Response · Authors · 2025-11-21
> **Official Comment by Authors (5/5)**
>
> ---
>
> > **Q7:Since mined text can include sensitive inputs (typed passwords, emails), how to deal with?**
>
> **A:** This is an important concern, and we appreciate the opportunity to clarify. Our data source consists of **publicly available screen‑recorded tutorial and demo videos**, but we do not record private user desktops or any non‑public content. Any text visible in the video (including emails or usernames) is already voluntarily disclosed by the video creator.
>
> To further address your concerns, we want to ensure that our processing and *released artifacts* do not amplify potential sensitive information:
>
> - For **training**, we only access text that is visible in the video frames or transcripts; we do not attempt to infer or reconstruct hidden content (e.g., masked password fields).
> - For **release**, our planned manifest focuses on action metadata (types, coordinates, timestamps, model‑generated rationales). Before releasing any textual metadata, we will apply open-sourced conservative filters such as [presidio](https://github.com/microsoft/presidio) to remove obvious sensitive patterns, such as email‑like strings or domains that appear to be personal identifiers. We do not intend to release any content from fields labeled as passwords or fields where the value is visually masked.
> - The CC‑licensed demo subset will undergo the same or stricter filtering.
>
> We have added a short discussion of these safeguards in the ethics statement section to make clear that we treat potential sensitive text seriously and design our release to avoid exposing or amplifying private information beyond what is already visible in the original public videos.
>
>
> ---
>
> > **Q8:Could you incorporate UI element detection (e.g., layout parsing) to normalize parameterization into element-centric actions rather than screen coordinates, improving cross-resolution robustness?**
>
> **A:** We appreciate this insightful suggestion. Element‑centric actions based on UI layout parsing are indeed an attractive way to improve cross‑resolution robustness and invariance to screen geometry. Our current choice of **coordinate‑based actions** is driven by two practical considerations:
>
> 1. **Coverage of fine‑grained interactions.** Many common operations are not naturally associated with a single discrete UI element, such as:
>
> - placing the text caret between two characters inside a text box,
> - dragging to select a substring within a larger text field,
> - clicking tiny micro‑controls such as the “×” on a browser tab or small icons inside complex toolbars.
>   In these cases, existing layout parsers (e.g., OmniParser) often fail to recover a sufficiently fine‑grained element, whereas coordinates can straightforwardly represent the intended action.
>
> 2. **Strong coordinate‑level grounding in modern VLMs.** As reported in the Qwen2.5‑VL paper, the underlying model already achieves strong performance on standard grounding benchmarks such as RefCOCO and GUI‑oriented datasets like ScreenSpot. In our experiments, we find that this coordinate‑level grounding transfers well across typical resolution changes, making coordinate‑based actions robust enough for many practical settings.
>
> We view element‑centric actions as **complementary** to our approach: our video‑to‑action pipeline could be extended to predict element identifiers instead of (or in addition to) coordinates once reliable layout parsers are available for the target platforms. However, given the current limitations in fine‑grained element detection, we focus in this work on a coordinate‑based representation that uniformly handles text editing, micro‑controls, and other interactions where element boundaries are ambiguous. We will clarify this rationale and the potential for element‑centric extensions in the revision.
>
> ------
>
> We thank you again for your valuable feedback, which has strengthened our work substantially. The OSWorld-G evaluation, expanded contamination analysis, and concrete data release commitments directly address your core concerns. We are happy to provide any further clarifications during the discussion period.

---

### Author Response · Authors · 2025-11-21
**General Comment to All Reviewers and the Area Chair**

We sincerely thank all reviewers and the area chair for their thoughtful, constructive feedback on our work. We are encouraged that reviewers agree on the novelty and importance of our goal: turning large-scale, unlabeled screen-capture videos into training-ready trajectories for computer-use agents.

Several reviewers recognized our key strengths:

- **Novel and timely approach** (`Rbqb`, `wmub`): First fully-automated pipeline converting unlabeled screen-capture videos into training-ready trajectories, addressing the data bottleneck for GUI agents
- **Strong empirical validation** (all reviewers): Clear improvements over SFT-only baselines on OSWorld-Verified and AgentNetBench with positive scaling curves
- **Practical impact and reproducibility** (`Rbqb`, `wmub`, `c9gy`): Comprehensive pipeline from collection to training, with open-source release of tools and data

At the same time, reviewers also raised several concerns, which we appreciate and believe we have fully addressed as follows:

1. **Data Release and Reproducibility**: We provide concrete commitments with specific deliverables: Video ID manifest with full action metadata, reconstruction scripts, and CC-licensed subset with direct trajectories
2. **Quality Validation** :
    - provided quantitative validation results in appendix: 81% of generated inner monologue samples meet pre-training quality criteria.
    - added OSWorld-G grounding evaluation to provide more training analyses.
    - visualized pretraining data distribution and provided a comprehensive contamination study.
3. **Additional Specific Clarifications**: Numerous clarifications requested by the reviewers, highlighted in blue in the revised version, have been added.

These updates strengthen our paper and directly address the core concerns while preserving the practical contributions of our work. We welcome further questions during the discussion period and thank all reviewers for their constructive feedback, which helped refine our claims, clarify limitations, and improve the overall presentation of VideoAgentTrek.

---

### Meta-Review · Area_Chair_yZQb · 2026-01-04

**Summary:**

This paper receives 8864, with only one negative review. The main concerns are about the robustness against noises, the release plan, and the generalisation. The inability of release all data and related resources is a main issue of this paper, I would encourage the authors to release as many deliverables as possible, which would be useful for the community.

**Reviewer Concerns:**

The reviewers' concerns are mainly about the implementation details, the generalization and the release plan. the authors' rebuttal address some concerns, but the inability of release would still be an issue. I would not reject this paper due to this issue, but I want to highlight that this would significantly reduce my willingness to reduce this paper.

**Reviewer Scores:**

I think the review with 4 may increase the score, so I think this paper would become with positive reviews.

---

### Decision · Program_Chairs · 2026-01-26

Accept (Poster)